

# How does grid-resolution modulate the topographic expression of geomorphic processes?

Stuart W. D. Grieve[1], Simon M. Mudd[1], David T. Milodowski[1], Fiona J. Clubb[1], and David J. Furbish[2]

[1]School of GeoSciences, University of Edinburgh, Drummond Street, Edinburgh EH8 9XP, UK
[2]Department of Earth and Environmental Sciences, Vanderbilt University, Nashville, TN, USA

*Correspondence to:* Stuart W. D. Grieve (s.grieve@ed.ac.uk)

**Abstract.** In many locations, our ability to study the processes which shape the Earth are greatly enhanced through the use of high resolution digital topographic data. However, although the availability of such datasets has markedly increased in recent years, many locations of significant geomorphic interest still do not have high resolution topographic data available. Here, we aim to constrain how well we can understand surface processes through topographic analysis performed on lower resolution data. We generate digital elevation models from point clouds at a range of grid sizes from 1 to 30 meters, which covers the range of widely used data resolutions available globally, at three locations in the United States. Using this data, the relationship between curvature and grid resolution is explored, alongside the estimation of the hillslope sediment transport coefficient ($D$, in $m^2$ $yr^{-1}$) for each landscape. Curvature, and consequently $D$, values are shown to be generally insensitive to grid resolution, particularly in landscapes with broad hilltops and valleys. Curvature distributions, however, become increasingly condensed around the mean, and theoretical considerations suggest caution should be used when extracting curvature from landscapes with sharp ridges. Two methods of extracting channels from topographic data are tested. A geometric method of channel extraction that finds channels by detecting threshold values of planform curvature is shown to perform well at resolutions up to 30 meters in all three landscapes. The landscape parameters of hillslope length and relief are both successfully extracted at the same range of resolutions. These parameters can be used to detect landscape transience and our results suggest that such work need not be confined to high resolution topographic data. A synthesis of the results presented in this work indicate that although high resolution (e.g., 1 m) topographic data does yield exciting possibilities for geomorphic research, many key parameters can be understood in lower resolution data, given careful consideration of how analyses are performed.





## 1 Introduction

Geomorphologists have always made use of topographic data, from initial qualitative observations
of surface morphology and its link to process (e.g., Gilbert, 1909) to directly measuring landscape
geometries from contour maps, constraining river dynamics and morphometric relationships (e.g.,
Horton, 1932; Schumm, 1956; Chorley, 1957). Further quantitative analyses of the Earth's surface
were facilitated through the advent of gridded topographic data. Work to generate Digital Elevation
Models (DEMs) from photogrammetry, contour maps and active remote sensing platforms (Yam-
aguchi et al., 1998; Wolock and McCabe, 2000; Rabus et al., 2003; Walker and Willgoose, 2006)
produced datasets at tens to thousands of meters grid resolution, along with geomorphic analyses
designed for such datasets (O'Callaghan and Mark, 1984; Tarboton et al., 1991; Montgomery and
Dietrich, 1994; Burbank et al., 1996; Tarboton, 1997). Algorithms have subsequently been devel-
oped which exploit the higher resolution topographic data now available, predominantly from Light
Detection And Ranging (LiDAR), which not only refined existing techniques (Passalacqua et al.,
2010; Pelletier, 2013; Clubb et al., 2014) but also allowed the study of hitherto unresolvable features
on landscapes (Tarolli and Dalla Fontana, 2009; Vianello et al., 2009; Roering et al., 2010; DiBiase
et al., 2012; Tarolli, 2014; Milodowski et al., 2015b).

Presently, LiDAR data coverage is predominantly focused around locations of particular scientific
interest or infrastructural importance, as can be seen on many LiDAR data portals (e.g., Krishnan
et al., 2011). It is unlikely that global LiDAR coverage can be achieved in the near future, leaving
the provision of commercially available 12 meter TanDEM-X data (Krieger et al., 2007) and freely
available 30 meter Shuttle Radar Topography Mission (SRTM) data (Rabus et al., 2003) as the best
available data options for many study sites.

As a consequence of this data availability it is crucial to understand the limitations of lower res-
olution data when performing topographic analysis for geomorphic research. Extracting channels
from topography is a common requirement of many analyses and it is expected that the accuracy of
extracted channel networks will be affected by increasing grid resolution (Orlandini et al., 2011).
Roering et al. (2007), Hurst et al. (2013b) and Grieve et al. (2016b) used measurements of hillslope
length and relief to identify signals of landscape transience. However, all such work was performed
on high resolution topography and the impact of grid resolution on these metrics is unknown. Roer-
ing et al. (2007) and Hurst et al. (2012) demonstrated that the curvature of ridgelines measured from
high resolution topography can be used as a proxy for erosion rates in soil mantled landscapes. This
observation has been used in many studies where cosmogenic radionuclide derived erosion rates are
unavailable (Pelletier et al., 2011; Hurst et al., 2013c, b; Grieve et al., 2016b). However, it can also
be used in locations with an independent constraint on erosion rates in order to quantify a sediment
transport coefficient that relates hillslope sediment flux to topographic gradient, which is set by the
material properties of soils (Furbish et al., 2009). Therefore, understanding the effect of grid resolu-



tion on the extraction of curvature is crucial in order to evaluate the applicability of calculating the

sediment transport coefficient from coarse resolution data.

Here, we grid topographic data at a range of resolutions in order to test the sensitivity of these techniques to increasing grid size, with the aim of placing constraints on the estimation of common geomorphic parameters when LiDAR topographic data are unavailable.

### 1.1  Previous work

It has long been recognized that the scale of topographic data used in an analysis or model will have an impact on the scale of the processes which can be measured (Vaze et al., 2010). It is intuitive that in order to measure the properties of hillslope processes the resolution of the data must be high enough that variations in hillslope form can be captured adequately. The resolution of topographic data defines the Nyquist frequency, given as $(2Res)^{-1}$ where $Res$ is the grid resolution of the dataset

(Warren et al., 2004). The inverse of this frequency yields the minimum wavelength resolvable from a given dataset. In the example of a 1 meter grid size, the smallest features that could be resolved would have a length scale of 2 meters. Recognizing this, many authors have attempted to quantify this uncertainty, aiming to answer the question: at what point does a dataset become unsuitable for a given analysis? (e.g., Quinn et al., 1991).

Many attempts to constrain the error content of topographic measurements have focused on comparisons between elevation values taken from differing resolution data products, often in conjunction with field survey data, with the aim of discriminating between DEM generation methods. Walker and Willgoose (2006) performed a comparison of DEMs generated using cartometric and photogrammetric methods against field surveyed elevation data. They demonstrated that at grid resolutions of 6.25,

12.5 and 25 meters the cartometric DEM produced less error than the photogrammetric DEM when compared to the field surveyed data, collected at 3.25 meter intervals.

The advent of LiDAR-derived topographic data provided a new technique, and increased the range of possible grid resolutions, to evaluate. Hodgson et al. (2003) assessed the quality of high resolution topographic data sourced from interferometry and LiDAR for a heavily vegetated catchment in

North Carolina. This analysis demonstrated that, under such conditions, the LiDAR-derived DEM outperformed the interferometric data in addition to both classes of USGS DEM product. However, concerns were raised about the overall accuracy of the LiDAR data with a requirement for improved methodologies to be developed to process multi-storey vegetation. Further work was carried out in North Carolina to constrain the minimum number of LiDAR returns required to generate a DEM at

a given grid size (Anderson et al., 2006). This work indicated that a 5 meter grid (the finest resolution used) required approximately 115 points per hectare, whereas at 30 meter grid resolution the requirement reduced to approximately 35 points per hectare.

Vaze et al. (2010) resampled a 1 meter LiDAR-derived DEM to a range of grid sizes up to 25 meters, and assessed the accuracy of elevation values for each of these resampled grids when compared



to a 1 meter resolution field survey. It was found that there was little variation in the distribution of elevation values between the resampled data sets. However, when the data was compared with 25 meter DEMs generated from topographic maps and contour generalization, there were considerable errors, supporting earlier author's conclusions that LiDAR-derived topographic data contains more useful geomorphic information than other methods of topographic data collection.

Topographic gradient (or slope) is one of the most fundamental topographic derivatives across the disparate disciplines which utilize topographic data. This measurement has been used in geomorphology (e.g., Burbank et al., 1996), ecology (e.g., Milodowski et al., 2015a), soil science (e.g., Nearing, 1997) and hydrology (e.g., Zhang and Montgomery, 1994). Wolock and McCabe (2000) endeavored to constrain the accuracy with which this parameter can be calculated as grid resolu-

tion is increased from 100 to 1000 meters and showed that as the grid size is increased, there is a clear reduction in the slope values produced for a landscape. Similar wide scale analysis has also been performed within the context of global hydrological analysis (e.g., Hutchinson and Dowling, 1991; Jenson, 1991), indicating that from meter to kilometer scale the reduction in quality of slope measurements is an issue which must be considered when working with topographic data.

Gao (1997) considered the accuracy of slope measurements at locations manually classified as valleys, peaks and ridges. They found an initially small increase in the error of slope measurements at intermediate resolutions (10 to 20 meters) and a much more rapid increase in error between 20 to 30 meters resolution, suggesting a threshold minimum resolution for analysis of these landforms. More recent work has considered how high resolution LiDAR data impacts the quality of slope mea-

surements. Vaze et al. (2010) demonstrated a similar trend to previous authors working in lower resolution data: as grid resolution is decreased from 1 to 25 meters, there is a considerable reduction in the slope values generated for a landscape. Warren et al. (2004) evaluated the reliability of slope measurements by contrasting 10 methods of gradient calculation against field measurements of topographic gradient. The error between DEM and field-derived slope measurements was shown

to increase with grid resolution (from 1 to 12 meters), resulting in the recommendation to increase data resolution wherever possible to decrease errors in topographic analysis.

Numerous authors have considered the impact of grid resolution on hydrological applications, which often require slope calculation as a fundamental processing step. It has been demonstrated across many landscapes and scales that as grid resolution is decreased the upslope contributing area

will increase and the local slope will decrease, which will have a significant impact on any hydrological analysis (Wolock and Price, 1994; Zhang and Montgomery, 1994; Wu et al., 2008). Similarly, from the perspective of modeling global scale sediment fluxes to the oceans, Larsen et al. (2014) noted that measurements of slope dropped logarithmically with increasing grid spacing, and failing to account for this may lead to a substantial underestimate of the contribution of steep, montane

regions.





Kenward et al. (2000) performed analyses on the accuracy of hydrological networks generated through photogrammetry and radar interferometry at 5 and 30 meters grid resolution respectively. Their error analysis was extended to consider the vertical errors generated both through the downsampling of the topographic data, as well as from the techniques used to capture the topographic

information. Predicted catchment runoff was up to 7% larger in the lower resolution datasets, considered to be driven by both the vertical errors and the reduction in spatial resolution increasing variables such as upslope drainage area.

Topographic Wetness Index (TWI), calculated as $ln(A/S)$ where $A$ is the specific upslope area and $S$ is the slope, is used as a single variable to compare the hydrological setting of differing parts

of the landscape, providing insight into factors including groundwater properties and overland flow rates. Sørensen and Seibert (2007) used LiDAR data to test the robustness of TWI calculations at spatial scales ranging from 5 to 50 meters, concluding that the most sensitive part of the TWI calculation were the specific upslope area measurements. This sensitivity resulted in significant variation in the TWI values across the range of resolutions tested.

The accuracy of channel network extraction from topographic data was tested by Murphy et al. (2008), who tested a 1 meter LiDAR DEM and a 10 meter photogrammatically generated DEM against a field mapped channel network in a catchment in Alberta, Canada. The 1 meter LiDAR derived channel network was found to be the best representation of the field mapped channel network, exceeding the quality of an additional channel network mapped by hand from aerial photographs.

However, as no intermediate datasets were tested it is not possible to understand at what resolution the degradation in channel network extraction quality occurs at for this location.

As models of agricultural soil loss depend heavily on topographic variables such as slope, work has been carried out to understand the influence of grid resolution on calculated rates of soil loss. Schoorl et al. (2000) tested data resolutions from 1 to 81 meters and demonstrated that in all cases,

rates of predicted soil loss increased with grid resolution. However, the rates of soil loss were also influenced by the type of flow routing utilized, with the multiple flow direction algorithm (e.g., Freeman, 1991; Quinn et al., 1991) proving most sensitive to resolution decreases. Work by Erskine et al. (2007) considering models of crop yields in Colorado, USA, demonstrated that on relatively flat surfaces, such as agricultural fields, the spatial resolution is less important than the vertical

accuracy when predicting crop yields, with significant errors being produced due to centimeter scale vertical displacements. Increasing the grid size from 5 to 30 meters had limited effect on the yield calculations.

Although considerable work has been carried out on the sensitivity of various factors to grid resolution, much of it has been focused on a specific application (e.g., Wolock and Price, 1994; Schoorl

et al., 2000; Erskine et al., 2007; Sørensen and Seibert, 2007) with few studies considering the impact of DEM grid resolution within a geomorphic context. Here we aim to extend existing methodologies to constrain the utility of low resolution data products across a suite of geomorphic analyses to



understand: (1) How hillslope length, topographic curvature and relief vary with grid resolution, (2) How best to extract channel networks in lower resolution datasets in order to minimize errors, and (3) If it is possible to estimate sediment transport coefficients from low resolution topographic data, where an independent constraint on erosion rate is available.

## 2 Theory and Methods

### 2.1 Generating topographic data

Previous studies that have explored the impact of changing grid resolution on topographic or geomorphic parameters have typically produced coarser resolution topographic data by downsampling the highest resolution data product available for their study sites (e.g., Thompson et al., 2001; Anderson et al., 2006; Claessens et al., 2005; Sørensen and Seibert, 2007). Work has been undertaken to understand the influence of various re-gridding schemes on topographic measurements (Wu et al., 2008), with focus placed upon understanding the use of downsampling high resolution data in order to facilitate computationally expensive analysis on larger spatial areas with minimal loss in data fidelity. However, as computational power increases, cost decreases and more efficient algorithms are developed (Tesfa et al., 2011; Qin and Zhan, 2012; Braun and Willett, 2013; Schwanghart and Scherler, 2014) the need to downsample data for computational convenience becomes reduced. Instead, it becomes more important to understand the limitations of available data products, to facilitate geomorphic analysis in locations where high resolution topographic data are not available. This is of particular importance in many studies of natural hazards (e.g., Saha et al., 2002; Carranza and Castro, 2006) where data quality is limited. It will also open geomorphic research up to communities which do not have the resources to acquire high resolution topographic data.

As a consequence of these constraints we have generated topographic data for our three study sites without downsampling or re-gridding high resolution data products, as is commonly performed (Thompson et al., 2001; Anderson et al., 2006; Claessens et al., 2005; Sørensen and Seibert, 2007). Instead we have followed established techniques to grid the processed LiDAR point cloud data provided by OpenTopography (http://www.OpenTopography.org) at a range of data resolutions which span from 1 meter, considered to be the limit of the Oregon Coast Range dataset by Grieve et al. (2016a) to 30 meters, which is equal to the grid resolution of the global SRTM dataset (Rabus et al., 2003), the Advanced Spaceborne Thermal Emission and Reflection Radiometer (ASTER) dataset (Yamaguchi et al., 1998) and in excess of the TanDEM-X dataset (Krieger et al., 2007) and as such should span the vast majority of grid resolutions used in modern geomorphic research. The error estimates of the raw point clouds used in this re-gridding process are provided by OpenTopography and can be found in Table 1.





The point clouds are gridded using Points2Grid, which employs a local binning algorithm, searching for points within a circular window of radius defined by Kim et al. (2006) as,

$$Radius = \lceil \sqrt{2} Res \rceil. \tag{1}$$

An inverse distance weighted averaging approach is then performed to assign an elevation value to each grid cell. This approach, which has been employed in previous studies (Grieve et al., 2016a, b), yields a reliable representation of the topographic surface, with few data gaps and a minimal amount of interpolation.

The topographic data used in this study has been gridded at 20 resolutions, and Figure 1 provides representative hillshades of a section of Santa Cruz Island, highlighting the degradation of topographic information as grid resolution is increased.

## 2.2 Measuring curvature from topography

Landscape curvature has long been recognized as a key geomorphic characteristic of landscapes, from Gilbert (1909)'s qualitative observations of hilltop convexity to more recent approaches to quantify landform curvature using digital topography (e.g., Schmidt et al., 2003; Hurst et al., 2012). However, unlike other key landscape properties such as gradient (Gao, 1997; Wolock and McCabe, 2000; Warren et al., 2004; Vaze et al., 2010), hydrology (Wolock and Price, 1994; Zhang and Montgomery, 1994; Murphy et al., 2008; Wu et al., 2008) or soil characteristics (Schoorl et al., 2000; Erskine et al., 2007), the influence of grid resolution on curvature has not been fully explored, particularly within a geomorphic context.

This is particularly important with the proliferation of high resolution topographic data from LiDAR, allowing the analysis of curvature at increasingly fine scales. Recent developments in channel extraction techniques (Lashermes et al., 2007; Passalacqua et al., 2010; Pelletier, 2013; Clubb et al., 2014) typically require the identification of topographic convergence in high resolution topography using a curvature threshold. Roering (2008) and Hurst et al. (2012) demonstrated that hilltop curvature scales with erosion rate and as such demonstrated the importance of accurately constraining the impact of grid resolution on this landscape parameter. Its importance is highlighted by an increasing number of studies using this relationship as a proxy for erosion rate (Pelletier et al., 2011; Hurst et al., 2013c, b; Grieve et al., 2016b). Hilltop curvature can also be used to constrain the sediment transport coefficient of a landscape where an independent constraint on erosion rate is available (Hurst et al., 2013c).

The measured curvature of a topographic surface depends on the orientation of the measurement. Here, we consider two common types of curvature, with the following definitions: (1) Total curvature ($C_{Total}$), the curvature of a surface calculated in 2 dimensions (Evans, 1980; Zevenbergen and Thorne, 1987; Moore et al., 1991); and (2) Tangential curvature ($C_{Tan}$), the curvature calcu-





lated normal to the slope gradient (Mitášová and Hofierka, 1993). These two measures are employed
to extract hilltop curvature and channel networks, respectively. However, these definitions vary be-
tween studies and software packages: see Schmidt et al. (2003) for a full review of the varying
nomenclature and definitions of curvature measurements used in the literature.

Work by Schmidt et al. (2003) utilized 10 meter resolution DEMs to evaluate the most accurate
method for calculating curvature from digital topographic data. It was concluded that curvature could
be most accurately calculated when a 9 term polynomial was fitted to the elevation surface, with the
caveat that this will only be effective where the data quality is high enough. In cases where the data
are of lower accuracy, Schmidt et al. (2003) recommended using quadratics to fit the elevation data.
This work was extended by Hurst et al. (2012) to consider if these patterns held for high resolution
topographic data, and it was found that fitting a 6 term quadratic or 9 term polynomial yielded similar
results. Therefore, Hurst et al. (2012) chose to use the 6 term quadratic to compute curvature. For
this study we also chose to use the 6 term quadratic in order to reduce computation time, and more
importantly, to provide more robust curvature values as the data quality is degraded to resolutions
below 10 meters (Schmidt et al., 2003).

We calculate curvature using a circular window passed across the landscape, with a radius defined
by identifying scaling breaks in the standard deviation and inter-quartile range of curvature calcu-
lated at increasing window sizes, consistent with the length scales of individual hillslopes (Lash-
ermes et al., 2007; Roering et al., 2010; Hurst et al., 2012; Grieve et al., 2016a, b). Consequently,
curvature measurements at the hillslope scale can only be considered at data resolutions high enough
to resolve individual hillslope features, considered here to be no more than 10 meters, based on the
window sizes identified for each landscape. A quadratic function of the form,

$$\zeta = ax^2 + by^2 + cxy + dx + ey + f, \tag{2}$$

is then fitted to the elevation values within the window by least squares regression (Evans, 1980),
where $\zeta$ is the elevation, $x$ and $y$ are horizontal coordinates and $a$ through $f$ are fitting coefficients.
The fitted coefficients of this polynomial can be used to calculate differing types of curvature:

$$C_{Total} = 2a + 2b, \tag{3}$$

and

$$C_{Tan} = \frac{2ae^2 - 2cde + 2bd^2}{(d^2 + e^2)\sqrt{(1 + d^2 + e^2)}}. \tag{4}$$

From the measure of $C_{Total}$ for every cell in a DEM, we can also extract a subset of curvature
values from the hilltops. The value of curvature at a hilltop ($C_{HT}$) can be readily evaluated if the



positions of the hilltops are known. To extract hilltops we follow Hurst et al. (2012) in defining a hilltop as the boundary between two drainage basins of the same stream order. These points in the landscape can be algorithmically extracted once a channel network is defined through the identification of points in the landscape where two channels of the same Strahler order meet, and the

identification of that point's upslope contributing area. Each of these areas defines a basin of a given order and by repeating this process across the range of Strahler orders found in the landscape, a network of hilltops can be defined. This network is then used to sample the curvature values at these locations to provide the $C_{HT}$ values across the landscape. To ensure consistency between $C_{HT}$ measurements at changing grid resolutions, the same channel network, generated using the geomet-

ric method described in Section 2.3 from 1 meter resolution data, is used as the basis of the hilltop extraction algorithm.

For our data on hilltop curvature, $C_{HT}$, hilltops with a gradient exceeding $0.4$ are excluded as Hurst et al. (2012) demonstrated that this gradient is the point at which $> 15\%$ of sediment transport is nonlinear. Under nonlinear sediment flux hilltop curvature scales nonlinearly with erosion rate

(Roering, 2008), and consequently cannot be used as a proxy for erosion rates. As hilltops have a convex form, their curvature should be negative, so as a final step any points identified as hilltops which have a positive curvature are excluded from further analysis.

## 2.3   Channel extraction

Extracting channel networks from digital topographic data remains a fundamental challenge for

many areas of topographic analysis. Without the ability to discriminate between fluvial and hillslope domains, it is not possible extract many topographic metrics such as hillslope length (Grieve et al., 2016a), mean basin slope (DiBiase et al., 2010) or hilltop curvature (Hurst et al., 2012), and the accuracy of each of these metrics will be influenced by the accuracy of the channel network extracted. At a more fundamental level, the ability to identify where channels initiate will facilitate better

understanding of the processes acting at the transition between diffusive (hillslope) and advective (fluvial) sediment transport (Perron et al., 2008a).

Many authors have made use of field mapped channel heads both as a basis for geomorphic analysis and as a method for evaluating channel extraction methods (Montgomery and Dietrich, 1989; Orlandini et al., 2011; Julian et al., 2012; Jefferson and McGee, 2013; Clubb et al., 2014). Prior to

the availability of high resolution topographic data, contributing area and slope-area scaling thresholds were commonly used to define the location of channel heads directly from DEMs (Mark, 1984; O'Callaghan and Mark, 1984; Montgomery and Dietrich, 1989; Tarboton et al., 1991; Dietrich et al., 1992, 1993). The influence of increasing grid size on such channel extraction methods was evaluated by Orlandini et al. (2011), who demonstrated a strong sensitivity in predicted channel head location

to grid resolution, suggesting that coarser resolution data may not be suitable for channel extraction



through an area threshold. We apply the method described by Orlandini et al. (2011) to quantify the accuracy of an extracted channel network, detailed in Section 2.4.

Several methods have been proposed to identify channel heads from high resolution topography. Typically these methods exploit the high resolution nature of topographic data to resolve morpho-
metric or process based signatures of channel initiation, or the transition between the hillslope and fluvial domain (Lashermes et al., 2007; Passalacqua et al., 2010; Pelletier, 2013; Clubb et al., 2014). Here we evaluate how two techniques, one geometric method built upon work by Pelletier (2013) and Passalacqua et al. (2010), and one process based method, the DrEICH algorithm, developed by Clubb et al. (2014), are influenced by increasing grid cell size.

The DrEICH method was selected for evaluation as the technique on which it is based has been shown to operate successfully in lower resolution data (Mudd et al., 2014). The DrEICH method makes use of $\chi$ analysis, performed by integrating drainage area along a river profile to facilitate comparisons between river profiles of differing drainage area, with fewer uncertainties than tradi- tional slope-area analysis (Royden et al., 2000; Perron and Royden, 2013). When plotting the $\chi$
value against elevation for a river profile, river channels will plot as linear segments, whereas hill- slopes will display nonlinear segments. The DrEICH algorithm identifies the transition between these linear and nonlinear segments as the best fit location of the channel head.

The geometric method, used by Grieve et al. (2016b), removes noise from the raw topographic data using a Wiener filter (Wiener, 1949), as recommended by Pelletier (2013). This smoothed topog-
raphy is then processed to identify channelized portions of the landscape using a tangential curvature threshold (e.g., Pelletier, 2013), selected using the deviation of the probability density function of curvature from a normal distribution on a quantile-quantile plot (e.g., Lashermes et al., 2007; Pas- salacqua et al., 2010). The identified areas of channelization are then combined into a contiguous channel network by employing a connected components algorithm (He et al., 2008) and thinned into
a final channel network skeleton using the algorithm of Zhang and Suen (1984).

Channels were extracted from the 5, 10, 20 and 30 meter DEMs generated in Section 2.1 using both of the channel extraction methodologies. Parameters required in the operation of each algorithm were selected based on values used in previous studies (Grieve et al., 2016a, b) and these values can be found in Appendix A.

## 2.4 Comparing channel networks

To assess the accuracy of the channel networks extracted using both methods, we employ two mea- sures of quality described by Orlandini et al. (2011). These measures operate on classifications of the predicted location of channel heads placing each channel head into one of three categories: true positives ($TP$); false positives ($FP$) and false negatives ($FN$). A $TP$ is where a predicted channel
head from low resolution data occupies the same spatial location as the channel head derived from 1 meter resolution topography. A $FP$ is where a predicted channel head is placed in a location where



there is no channel head in the high resolution data. A $FN$ is when a channel head from high resolution topography does not have a predicted channel head from low resolution topography in the same spatial location.

We follow Orlandini et al. (2011) in employing a 30 meter search radius around the 1 meter-derived channel heads, and consider a low resolution channel head falling within this radius to be spatially coincident. This has the effect of normalizing the size of each channel head point, to ensure that we can perform comparisons between predictions made at different spatial resolutions.

The reliability, $r$, of a channel extraction method is the ability of a method to not predict channel

heads in areas where none are located and is calculated as,

$$r = \frac{\sum TP}{\sum TP + \sum FP}, \tag{5}$$

where $\sum TP$ is the total number of true positives and $\sum FP$ is the total number of false positives. The sensitivity, $s$, of a channel extraction methodology is given by,

$$s = \frac{\sum TP}{\sum TP + \sum FN}, \tag{6}$$

where $\sum FN$ is the total number of false negatives. The sensitivity is the ability of a method to predict all of the channel heads expected. Using these two indexes it is possible to quantify the quality of channel heads predicted using low resolution data, as well as understand why a particular method fails, by distinguishing between methods which fail due to either over or under predicting the number of channel heads in a landscape, or by simply placing channel heads in the wrong spatial

location.

**2.5 Estimating the hillslope sediment transport coefficient from hilltop curvature**

The sediment transport coefficient, $D$ $[L^2 T^{-1}]$ (dimensions of [M]ass, [L]ength and [T]ime denoted in square brackets), of a landscape is a measure of its sediment transport efficiency and was demonstrated by Furbish et al. (2009) to be controlled by the material properties of soil such as grainsize,

cohesion and thickness. The value of $D$ within a landscape will exert a control on the morphology of hillslopes (e.g., Roering et al., 1999). Diffusion-like hillslope evolution can be modeled using a 1D mass conservation equation, assuming that the contribution to surface lowering from chemical processes is negligible when contrasted with the signal of surface lowering from physical processes (e.g., Roering et al., 1999; Mudd and Furbish, 2004),

$$\rho_s \frac{\partial \zeta}{\partial t} = -\rho_s \frac{\partial q_s}{\partial x} + \rho_r U, \tag{7}$$





where $\zeta$ [L] is the elevation of the land surface, $\rho_s$ and $\rho_r$ $[ML^{-3}]$ are densities of dry soil and rock, respectively and $U$ $[LT^{-1}]$ is the uplift rate. In steady state landscapes, where $U = E$ and $\partial z/\partial t = 0$, Equation 7 simplifies to,

$$\frac{\rho_r}{\rho_s} E = \frac{\partial q_s}{\partial x}, \tag{8}$$

with $E$ $[LT^{-1}]$ denoting the erosion rate. To solve this equation, a statement of the volumetric sediment flux per unit contour length, $q_s$ $[L^2T^{-1}]$, must be derived. A nonlinear relationship between sediment flux and topographic gradient has been proposed by a number of authors (Andrews and Bucknam, 1987; Koons, 1989; Anderson, 1994; Howard, 1997; Roering et al., 1999, 2001; Pelletier and Cline, 2007). Support for such models has been found from both tests of the resulting topo-

graphic predictions (Roering et al., 2007; Hurst et al., 2012; Grieve et al., 2016a), as well as through independent measurements of sediment flux across hillslopes (Roering et al., 2001; Roering, 2008).

The nonlinear model proposed by Andrews and Bucknam (1987) and Roering et al. (1999) is of the form,

$$q_s = DS \left[ 1 - \left( \frac{|S|}{S_c} \right)^2 \right]^{-1}, \tag{9}$$

where $S_c$ is a critical gradient, and as the hillslope gradient approaches this threshold, $q_s$ asymptotes towards infinity.

At low hillslope gradients (e.g. on hilltops), the term within brackets in Equation 9 approximates to unity (Hurst et al., 2012). Equation 9 can therefore be substituted into Equation 8 and can be solved for $D$ on low gradient hilltops, assuming that an independent constraint on $E$ is available,

$$D = -\frac{E\rho_r}{C_{HT}\rho_s}. \tag{10}$$

**2.6  Hillslope length and relief**

Hillslope length ($L_H$) is a crucial landscape parameter to constrain as it controls the rate of mass flux by overland flow within catchments (Dunne et al., 1991; Thompson et al., 2010; Dunne et al., 2016), influences rates of soil erosion (Liu et al., 2000), and presents a first order control on the

maximum source area of landslides (Hurst et al., 2013a). Furthermore, it may be used to demonstrate nonlinearity in hillslope sediment flux (Roering et al., 1999, 2007; Grieve et al., 2016a, b).

Many studies have attempted to calculate hillslope length through the inversion of drainage density (Tucker et al., 2001), analysis of plots of local slope against drainage area (Roering et al., 2007), direct measurements from topographic maps (Hovius, 1996; Talling et al., 1997), and by measuring

the length of overland flow from ridgeline to channel (Hurst et al., 2012; Grieve et al., 2016a). Grieve



et al. (2016a) demonstrated that the most geomorphologically suitable technique to use, particularly in the context of hillslope sediment transport, was that of measuring the length of overland flow. An additional measure which can be derived from this technique is the topographic relief, which is the difference in elevation between a hilltop and channel connected by a hillslope flow path.

Topographic relief has been estimated in a number of ways and is frequently used in studies of tectonic geomorphology (e.g., Gabet et al., 2004; Hilley and Arrowsmith, 2008; Gallen et al., 2011, 2013). Furthermore, topographic relief may be used to generate dimensionless erosion and relief plots (Roering et al., 2007; Hurst et al., 2012; Sweeney et al., 2015; Grieve et al., 2016b), which can be used to identify landscape transience (Hurst et al., 2013b; Mudd, 2016). Consequently, we intend

to test the robustness of measuring hillslope length and relief as grid resolution decreases, with the aim of facilitating increased confidence in geomorphic analyses performed in locations where high resolution topography is unavailable.

Using the 20 topographic datasets generated in Section 2.1 for each of the three landscapes, hillslope length measurements were generated following the methods outlined in Grieve et al. (2016a).

We measured hillslope length on each dataset using two different channel networks. Firstly, channel heads were extracted from the highest resolution data set, in each case 1 meter, using the geometric method outlined in Section 2.3. These high resolution channel heads were mapped onto the coarser resolution topographic data, to ensure that changing channel extraction results will not have an influence on the measures of hillslope length. This allows improved isolation of the factors driving

variations in hillslope length as grid resolution is decreased. Secondly, the analysis was performed using coarser resolution channel networks extracted using the geometric method of channel extraction. We use the geometric method as opposed to the DrEICH method because, as we will show below, the geometric method is less sensitive to grid resolution. These two channel networks effectively provide upper and lower bounds on the accuracy of hillslope length and relief measurements.

**3 Study sites**

Three study sites from the United States have been selected for this study: Santa Cruz Island, California; Gabilan Mesa, California; and Oregon Coast Range, Oregon. The first two sites have regularly spaced valleys at a range of length scales, particularly Gabilan Mesa, which has been the focus of previous work in this context (Perron et al., 2008b, 2009). Santa Cruz Island, while less studied in

the context of topographic analysis than the Gabilan Mesa, has a wider range of hilltop curvatures (Figure 2). The Oregon Coast Range has been considered to be very regular, with uniform first order drainage areas (Roering et al., 1999, 2007). However, more recent work has demonstrated the spatial variability of many topographic measurements in this landscape (Marshall and Roering, 2014; Grieve et al., 2016b) and as such provides a more challenging test case for our analyses. Further-

more, these sites were selected as they each have high resolution LiDAR data covering a large spatial





area, and have been the subject of many previous studies (Reneau and Dietrich, 1991; Roering et al., 1999, 2001; Montgomery, 2001; Pinter and Vestal, 2005; Roering et al., 2007; Perron et al., 2009; Perroy et al., 2010, 2012; Marshall and Roering, 2014; Grieve et al., 2016a, b) which should provide a good basis for the evaluation of the results of this study in a wider geomorphic context.

### 3.1 Gabilan Mesa

Gabilan Mesa, a section of the Central Coast Ranges in California, USA (Figure 2b) is a highly regular landscape with very gentle transitions between hillslopes and channels, which correspond to topographic predictions of diffusion-like sediment transport (Roering et al., 2007). The area's semi-arid climate supports range of vegetation from oak savanna to chaparral shrubland (Shreve, 1927; Roering et al., 2007). The nature of this lower density vegetation allows a larger proportion of LiDAR pulses to reach the ground, requiring less processing and interpolation to generate a final bare earth DEM for analysis (Liu, 2008; Meng et al., 2010).

A series of large, linear canyons running north east to south west are fed by parallel tributaries which flow perpendicular to the main trunk channel. These regularly spaced valleys present two distinct length scales in the landscape which have been observed both qualitatively (Dohrenwend, 1978, 1979) and quantitatively through measurements of hillslope length distributions (Grieve et al., 2016a). Relationships between dimensionless erosion rate and relief, the uniformity of hilltop curvatures, and the regularity of valley spacing have all been used to assert that this landscape is in steady state (Roering et al., 2007; Perron et al., 2009; Grieve et al., 2016b).

### 3.2 Santa Cruz Island

Santa Cruz Island (Figure 2c), the largest of the eight California Channel Islands located to the west of California, USA, is divided by a large east-west trending valley, which follows the Santa Cruz fault (Pinter et al., 2003; Muhs et al., 2014). Parallel to this valley are two large ridges, one to the north and one to the south, which exhibit regularly spaced parallel channels draining north to south (Pinter et al., 1998; Pinter and Vestal, 2005); this regular pattern is particularly evident in the northwest section of the study area. The Santa Cruz Fault has been demonstrated to have left-lateral strike slip motion, which deflects channels away from perpendicular to the main valley in the center of the island (Pinter et al., 1998). Studies of marine terraces in the region suggest that the Channel Islands have been steadily uplifted through the late Quaternary (Muhs et al., 2014).

The island has a Mediterranean climate similar to that of Gabilan Mesa (Pinter and Vestal, 2005), supporting extensive grassland with occasional patches of pine forest and chaparral vegetation (Pinter and Vestal, 2005; Perroy et al., 2010, 2012). Human activities led to overgrazing across the island at the turn of the 19th century, causing a period of gullying and rapid erosion, particularly evident in the southwest of the island (Pinter and Vestal, 2005; Perroy et al., 2012). The LiDAR data collected for this location has been extensively tested and ground truthed, ensuring that it is suitable for use in





a geomorphic context (Perroy et al., 2010) and for performing topographic analysis at high spatial resolutions.

### 3.3 Oregon Coast Range

The Oregon Coast Range in Oregon (Figure 2d), USA, is a densely vegetated upland landscape,
dominated by coniferous and hardwood forests (Schmidt et al., 2001), with a humid climate (Roering et al., 1999). Qualitative observations of the landscape suggest that the valleys are regularly spaced, with a particular uniformity found in the dimensions of first order drainage basins (Roering et al., 1999, 2007; Marshall and Roering, 2014). Such observations have been supported by measurements of hillslope length across the landscape (Grieve et al., 2016a). However, comparisons
of the dimensionless relief and erosion rate performed by Grieve et al. (2016b) highlight the small scale topographic variability inherent in this otherwise regular landscape. The Oregon Coast Range is considered to be in steady state due to the correlation between uplift rates from marine terrace data (Kelsey et al., 1996) and erosion rates from cosmogenic radionuclides (Beschta, 1978; Reneau and Dietrich, 1991; Bierman et al., 2001; Heimsath et al., 2001). The hillslopes are steeper and the
ridgelines sharper than in Gabilan Mesa, consistent with observations of debris flows and shallow landsliding across the range (Dietrich and Dunne, 1978; Heimsath et al., 2001; Montgomery, 2001), which have the potential to create a distinct topographic signature (Booth et al., 2009).

## 4 Results

### 4.1 Curvature

Figure 3 illustrates the variations in total curvature with grid resolution for a section of Santa Cruz Island. As the cell size is increased the range of $C_{Total}$ measurements are reduced, with much of the landscape becoming apparently planar. Within the black box, which covers the same spatial area as the boxes in Figure 1, the impact of degrading resolution on small topographic features is observed, with the curvature signal of this first order feature being lost as the grid size approaches 30 meters.

Figure 4 displays the variations in the distribution of total and tangential curvature measurements with grid resolution for each of the study landscapes. Santa Cruz Island shows little variation in mean and median curvature with resolution, with the majority of the changes in each distribution with resolution occurring at the extremes of the curvature distribution for each dataset, as the representation of ridgelines and channel bottoms becomes increasingly diffuse. As resolution is decreased, the range
between 2nd and 98th and the 1st and 3rd quartiles decreases, with a more rapid reduction in the more extreme values than in the quartiles (Figure 5). While this effect is most marked at the extremes, the distributions are condensed across all percentile intervals as grid resolution is increased beyond 3-4 meters. This behavior is observed for both $C_{Total}$ and $C_{Tan}$ as grid size is increased.



In the Oregon Coast Range for both measurements of curvature there is little variation between the

1, 2 and 3 meter datasets, with a broad range of measurements shown in the probability distributions. Beyond this point the mean and median do not significantly change, but as in Santa Cruz Island, the overall distribution of measurements compresses towards the average value for the landscape. The Gabilan Mesa data show similar trends to that of Santa Cruz Island, but exhibit less variability at lower resolutions. The probability distributions of each measurement also exhibit less change

with resolution than the other two datasets, indicating a reduced sensitivity to grid resolution at this location.

### 4.2  Channel networks

Figure 6 provides a qualitative overview of the changes of channel network extent with decreasing grid resolution for both methods, across the three test landscapes. In each case the general patterns

are that as the grid resolution is decreased, the lowest order channels are lost, as they exist at a spatial scale below that of the data resolution. In contrast, much of the predicted networks appear to occupy similar spatial locations in larger, higher order channels where the topographic signal of a channel is more pronounced. The geometric method shows less reduction in drainage density than the DrEICH method, as data resolution is decreased.

Figure 7 provides a quantitative assessment of channel extraction quality by presenting the indexes of reliability and sensitivity for both the geometric channel extraction and extraction based on DrE-ICH, as the grid size is increased. In Gabilan Mesa the channels extracted by the geometric method exhibit a high reliability which does not decrease considerably with increasing grid size, suggesting that for each resolution step a large proportion of the predicted channel heads are spatially coincident

with the channel heads generated from the 1 meter data. The sensitivity values for this method and location are lower, and decline more steadily with decreasing grid resolution, suggesting an increasing number of channel heads being missed by the algorithm as grid size is increased. The DrEICH method does not perform as well in Gabilan Mesa, with lower index values for the 5 meter data than the geometric method, and a rapid decline towards index values of 0, suggesting that the predicted

channel heads bear little relation to the channel heads from the 1 meter data.

In Santa Cruz Island the geometric method's reliability index is similar to Gabilan Mesa, however the sensitivity index is not as high, which indicates a large number of channel heads are being missed but where a prediction is made it is typically accurate. The DrEICH method exhibits a similarly large reliability initially, but again shows more rapid degradation in the index value as grid size is

increased. The sensitivity values again decline more rapidly and reach a 0 value at 20 meter grid resolution.

The data for the Oregon Coast Range shows similar patterns for both methods, although the geometric method exhibits systematically larger index values. In each case the reliability increases slightly from 5 to 10 meter resolution and then declines gradually towards 30 meter grid size. The





sensitivity indexes for both methods begin at a larger value than the reliability indexes, and steadily
      decline towards 0. A sensitivity value exceeding the reliability value suggests that in this landscape
      there are fewer missed channel heads in the 5 meter data, but at the expense of too many predicted
      channel heads in locations where there are none predicted in the 1 meter data.

### 4.3   Sediment transport coefficient

Using the values for hilltop curvature generated in Section 4.1, published parameters for erosion
      rate and material properties outlined in Table 2 and Equation 10, the average sediment transport
      coefficient ($D$) of each landscape can be calculated as a function of grid resolution. Figure 8 displays
      the relationship between diffusivity and grid resolution for each of the three study sites. The data for
      Santa Cruz Island and Oregon Coast Range both show a gradual increase in diffusivity with grid

size, the rate of which reduces with increasing grid size. The Gabilan Mesa data does not exhibit
      the same trend, with little variability in calculated $D$ values as resolution is decreased. Although the
      Oregon Coast Range and Santa Cruz Island datasets exhibit an increase in estimated $D$, all of the
      values for each location fall within the range of values for $D$ compiled by Hurst et al. (2013c).

### 4.4   Hillslope length and relief

The hillslope length measurements for Santa Cruz Island calculated using 1 meter channel heads
      (Figure 9a) show little variation in the distribution of the data up to 10 meter resolution, with the
      main difference being the increase with grid size in the $2^{nd}$ percentile measurements, which is a
      trend observed within each of the datasets. The mean and median values also gradually decrease
      towards the 10 meter resolution dataset, before gradually increasing towards the 30 meter resolution

step. However, these variations are very small, with the overall distributions of hillslope length and
      relief not varying considerably between resolution steps. When the same hillslope length algorithm
      is applied using channel networks extracted using the geometric method for each resolution step
      (Figure 9c), there is little change in the distribution or average values of $L_H$ until beyond the 10
      meter resolution step. Beyond this point the measurements of hillslope length are clearly affected by

the reduction in accuracy of the channel network. The relief measurements for both channel head
      methods (Figure 9b, d) in Santa Cruz Island exhibit little resolution dependence up to 10 meter cell
      sizes, beyond which point the values increase steadily. In the case of the 1 meter channel heads the
      distribution becomes compressed around the average values at lower resolutions, whereas with the
      variable channel head dataset the distribution of values increases with decreasing resolution.

565       In Gabilan Mesa the hillslope length measurements calculated using 1 meter channel heads (Fig-
      ure 10a) show a gradual reduction in mean and median values between the highest resolution data
      and the 8 meter resolution data before a small plateau and then a small increase until the 30 meter
      dataset. The average relief values calculated for the same dataset increase steadily by approximately
      20 meters between the highest and lowest resolution datasets (Figure 10b). The distribution of relief



measurements are broadly consistent between 1 and 5 meter resolutions before reducing about the
median as grid size is increased. The same trends are apparent in the hillslope length and relief data
calculated using the variable channel heads (Figure 10c, d) with little change between the two pairs
of datasets.

The hillslope length measurements for the Oregon Coast Range with channel heads from the 1
meter data (Figure 11a) again show a gradual reduction in the median values with a gradual increase
in the mean values until 20 meter grid resolution. Beyond this point the data becomes considerably
more variable, with a large increase in both the mean and median results. The relief data shown
in Figure 11b is the most consistent of the three landscapes with very little variation in the values
until they begin increasing with cell size at approximately 20 meter resolution. The data presented
in Figure 11c and d shows the most sensitivity to grid resolution of the three landscapes. Average
hillslope length values reduce towards 10 meters before stabilizing and then rapidly increasing in
the same manner as the fixed channel head data. The relief measurements show a gradual decline in
mean relief across the range of resolutions from 1 to 10 meters, where the fixed data shows much
less variation.

## 5   Discussion

### 5.1   Curvature and the problem of resolution-dependent filtering

Across the three landscapes the variance of the distributions of both total and tangential curvature
values are systematically reduced as resolution is decreased, an effect that is particularly notable af-
ter the grid resolution exceeds 3-4 meters (Figure 4). In each of the three datasets, the inter-quartile
ranges remain relatively constant, whereas beyond 4 meters resolution in each case the range be-
tween the 2nd and 98th percentiles reduces rapidly (Figure 5), demonstrating that the majority of the
loss of curvature information occurs at the extremes of the distribution.

In producing a DEM, we are sampling a complex two-dimensional elevation signal, in which
spatial variations in geomorphic process drive variations in topographic amplitude at different wave-
lengths (Perron et al., 2008b). Decreasing the grid resolution of DEMs acts as a low-pass filter on
this topographic signal, which preferentially degrades features in the topography that have signif-
icant amplitude at small wavelengths, such as sharp ridgelines, narrow valley bottoms, and local
topographic roughness generated by, for example, landslides, tree throw and rock exposure (Fig-
ures 1 and 3). While the position of ridges and valleys is preserved in coarser resolution data, the
magnitude of their associated curvature values is reduced as cell sizes increase; this effect is par-
ticularly marked for hillslopes in which curvature is focused at the ridge crest and valley bottoms,
a common characteristic of more rapidly eroding landscapes (Roering et al., 1999, 2007). For first
order landscape features, such as gullies, landslide scars and first order channels, increasing grid





size eventually results in the complete loss of topographic information, as highlighted in Figure 1
and Figure 3.

### 5.1.1   Topographic filtering and its implications for curvature and slope measurements

To interpret the observed loss of fidelity that accompanies coarser grids evident in Figures 4 and 5,
it is illustrative to examine the spectral behavior of a simplified 1 D system. We acknowledge that a
1 D approach cannot fully describe complex two dimensional topography of real landscapes, but a
one dimensional system is amenable to mathematical treatment that can at least give us qualitative
insight into trends observed in our data. In addition, some of the features of interest, for example
ridgelines and channels, can be roughly approximated as one dimensional structures within a two
dimensional landscape.

Curvature in one dimension, $C_x$ [L$^{-1}$], is often approximated with the differencing equation:

$$C_x = \frac{\zeta_{(x-\Delta x)} - 2\zeta_x + \zeta_{(x+\Delta x)}}{(\Delta x)^2}, \tag{11}$$

where $\zeta$ [L] is the elevation of the land surface, $x$ [L] is a location in space, $C_x$ is the curvature
at location $x$, and $\Delta x$ [L] is the grid interval. The subscripts denote the discrete locations where
elevation is evaluated. We can calculate the wavenumber response function ($H(\omega; \Delta x)$) from this
filter (Jenkins and Watts, 1968):

$$H(\omega; \Delta x) = \frac{2}{(\Delta x)^2}[cos(\omega \Delta x) - 1), \tag{12}$$

where $\omega$ = 2 $\pi/L$ [L$^{-1}$] is the wavenumber with wavelength $L$ [L] (that is, higher wavenumbers
correspond to higher frequencies and thus shorter wavelengths). Using this function, we can calcu-
late the gain, $G(\omega; \Delta x)$, which is the amplitude of the filtered signal (in this case curvature) to the
amplitude of the original signal (in this case elevation) at the wavenumber $\omega$. The theoretical gain
for continuous waveforms (i.e., not discrete filters like Equation 11) is $\omega^2$. The gain of a discrete
filter is the modulus of the wavenumber response function (Jenkins and Watts, 1968), so in the case
of Equation 12 the resultant gain, $G(\omega; \Delta x)$ is:

$$G(\omega; \Delta x) = \frac{2}{(\Delta x)^2}[1 - 2cos(\omega \Delta x) + cos^2(\omega \Delta x)]^{1/2}. \tag{13}$$

In the case of our curvature filter (Equation 11), the gain function reveals how high frequency
waveforms (e.g., ridgecrests, treethrow mounds, local roughness) in the elevation data involve rel-
atively large values of curvature, whereas low frequency elevation waveforms (e.g., ridge-valley
features or geologic folds) with the same amplitude involve relatively small curvatures. Crucially,
however, the discrete filter does not retain all of the high frequency information. We can calculate





what information is lost by calculating the fidelity, which is the ratio between discrete gain (Equa-
tion 13) and the theoretical gain ($\omega^2$):

$$F(\omega; \Delta x) = \frac{2}{(\Delta x)^2 \omega^2} [1 - 2cos(\omega \Delta x) + cos^2(\omega \Delta x)]^{1/2}. \tag{14}$$

Fidelity is a function of the ratio between the grid interval and the wavelength (Figure 12); when
the fidelity is unity, the discrete filter exactly reproduces the underlying continuous function. As
the frequency approaches the Nyquist wavenumber ($L/\Delta x = 1/2$), fidelity decreases; a fidelity of
only approximately 0.4 is achieved at the Nyquist wavenumber itself. To achieve a fidelity, $F$, of
0.9 requires that $L/\Delta x$ is equal to approximately 6 grid points per wavelength. A fidelity $F = 0.95$
requires 8 points per wavelength, and $F = 0.99$ requires 18. Therefore, while the grid resolution im-
poses a minimum wavelength that can be resolved (defining the Nyquist wavenumber), the behavior
of the fidelity function (Figure 12), clearly illustrates that curvature information will be lost when
calculated for features with wavelengths greater than, but still close to the minimum resolvable at the
Nyquist wavenumber. In this simple example, using 1 meter resolution data, one could only capture
the curvature of a one dimensional ridgeline that had a wavelength of 3-4 meters (one does not need
the entire wave to capture the peak of the waveform), but with loss of fidelity on the magnitude of
the curvature (i.e., underestimating its value).

In our study we have not computed how topographic gradient varies as a function of grid resolution
because this has been examined by many previous authors (e.g., Gao, 1997; Warren et al., 2004;
Vaze et al., 2010). However our treatment of the properties of a one dimensional filter can give some
insight into previous results. Consider a simple central-difference approximation of the topographic
gradient ($S_x$, dimensionless):

$$S_x = \frac{\zeta_{(x+\Delta x)} - \zeta_{(x-\Delta x)}}{2\Delta x}. \tag{15}$$

We can follow the same series of operations that we performed on Equation (11) to arrive at the
fidelity of Equation (15), denoted as $F_S$, taking into account that the theoretical gain is $\omega$:

$$F_S(\omega; \Delta x) = \frac{1}{\Delta x \omega} [sin(\omega \Delta x)]. \tag{16}$$

The form of Equation (16) (Figure 12) formally illustrates why estimates of slope tend to system-
atically decrease with increasing grid interval $\Delta x$. Namely, an increasing $\Delta x$ is able to resolve less
local (high wavenumber) elevation structure while picking out the slope of more regional structure.
The fidelity increases as the ratio of the grid interval to the wavelength, $L/\Delta x$, decreases (Fig-
ure 12). To achieve a fidelity $F_S = 0.9$, for example, requires $L/\Delta x$ or approximately 8 grid points
per wavelength. A fidelity $F_S = 0.95$ requires 11 points per wavelength, and $F_S = 0.99$ requires 18.



The fidelity of the one dimensional gradient operator goes to zero when approaching the Nyquist wavenumber ($L/\Delta x = 1/2$). These results explain the pronounced loss of gradient information in coarse resolution data observed by many authors (e.g., Gao, 1997; Warren et al., 2004; Vaze et al., 2010).

### 5.1.2    Total and tangential curvature

Having explored simplified one dimensional filters, we now return to our two dimensional results. Although real landscapes are two dimensional and we use polynomial fitting rather than simple differencing as in Equation 11, we can still use Equation 14 as a qualitative indicator of the grid spacing required for appropriate curvature estimates. In the Gabilan Mesa, where ridgelines are broad, lower resolution data can still capture the curvature with relatively high fidelity. However, in

locations with sharper ridgelines, such as Santa Cruz Island, the narrowest ridgelines are no longer adequately resolved as the grid size is increased, as can be seen in Figure 3.

    The loss of fidelity predicted by the simple one dimensional system (Equation 14) qualitatively predicts the pattern observed in Figures (4 and 5), namely that the curvature values are smeared over a greater length-scale leading to apparently broader ridges with resolution, and a systematic under-

estimation of their peak elevations. This highlights that in conjunction with data quality, landscape morphology also exerts a control on the optimal resolution to use for a given study, where landscapes with more gradual hillslope to valley transition morphologies can be analyzed using coarser resolution topographic data with more confidence.

    Santa Cruz Island and the Oregon Coast Range have the highest tangential curvature at 1 meter

resolution. High tangential curvature at Santa Cruz Island corresponds to observations of extensive gullying and hillslope erosion (Pinter and Vestal, 2005; Perroy et al., 2012). In the Orgeon Coast Range, features such as pit and mound topography produced by tree throw and other biotic activity are resolved in the LiDAR dataset (Roering et al., 2010; Marshall and Roering, 2014), which manifest as an increase in values of curvature. However this could also be indicative of non-topographic

noise in the DEM surface produced during the processing of the point clouds, which is particularly required in heavily forested locations (Liu, 2008; Meng et al., 2010) such as the Oregon Coast Range. This suggests an unfortunate collinearity between the two causes of small wavelength topographic noise and warrants further testing in future to disentangle synthetic and natural noise from high resolution topographic measurements. However, high curvature is not solely a manifestation of stochastic

disturbance on local topographic roughness, but is also generated at narrow valley bottoms, and at ridgelines where erosion rates are rapid relative to the hillslope sediment transport coefficient (Roering et al., 2007; Hurst et al., 2012). Gabilan Mesa exhibits much lower curvature values than the other two locations, which is a consequence of high landscape diffusivity, indicating that sediment transport at Gabilan Mesa is dominated by diffusion-like processes (Roering et al., 2007), smoothing

the landscape and reducing the tangential curvature of the hillslope surface.



### 5.2 Channel extraction

It is intuitive to consider that when extracting channel networks at any data resolution, regardless of method, the higher order, larger channels will be more accurately constrained than lower order channels. This pattern is observed in each of the study landscapes, with the majority of the variations
in channel locations occurring in first and second order channels. Such loss of low order channels from datasets has implications for studies focusing on upland areas, in particular where detailed measurements which depend on channel network position are performed.

The contrast between the extent of channel networks and their indexes of quality for the two methods outline that a geometric method of channel extraction outperforms the process-based DrEICH
algorithm. Due to the relative simplicity of the geometric method of channel extraction, errors inherent in the DEM are not compounded at the same scale as the DrEICH algorithm, which performs more operations on topographic data. As the geometric method identifies channels based on their tangential curvature, although channel head features may be smoothed out of the DEM as cell size is increased, the channel will still express some positive curvature in lower resolution data. The ini-
tiation point may be located downslope of the true channel head, but even in this worst case most of the channel network will be extracted correctly. This is observed in Figure 6 which shows a gradual reduction in drainage density as the grid resolution is increased.

The indexes of quality defined by Orlandini et al. (2011) provide a clear framework to understand the quality of channel head predictions using these two methods as data resolution is decreased.
In each case, the geometric method outperforms the DrEICH method, both in the accuracy of the channel heads which are predicted, and in the ability of the method to not predict channel heads in locations where no channel exists. These indexes are influenced by the size of the search radius around each channel head, and reducing this radius would decrease the index values. However, the use of a 30 meter search radius allows comparisons to be drawn between predictions made at
different data resolutions, and also between this study and that of Orlandini et al. (2011).

This assessment of high resolution methods on degraded quality data demonstrates the ongoing challenges that channel extraction poses to the geomorphology community. Orlandini et al. (2011) performed extensive testing on channel extraction using threshold channel extraction methods, and demonstrated similar limitations when channels were extracted using lower resolution data. Our re-
sults suggest that a geometric method of channel extraction will provide an optimal channel network as data quality is reduced, particularly in uniform landscapes such as Gabilan Mesa. However, the only way to ensure the highest quality results is to employ high resolution data in conjunction with field mapping of channel network extents.





### 5.3 Sediment transport coefficient

The predicted values of the sediment transport coefficient ($D$) for the 1 meter data fall within the range of values compiled by Hurst et al. (2013c), and estimated for the Oregon Coast Range and Gabilan Mesa by Roering et al. (1999) and Roering et al. (2007). This suggests that this method can produce useful estimates of $D$ when employing high resolution topography.

The sediment transport coefficients calculated at the Oregon Coast Range and Santa Cruz Island 740 locations both increase with grid resolution, reflecting the sensitivity of $C_{HT}$ to grid resolution in each of these locations. Despite the Oregon Coast Range eroding 45% more rapidly (Table 2) than Santa Cruz Island, the rate of increase in $D$ measurements remains similar between the two landscapes. Gabilan Mesa data are generally insensitive to an increase in grid cell size, as the scale of hilltop widths measured in Gabilan Mesa is on the order of tens of meters. This allows datasets 745 with cell sizes approaching half the width of a hilltop to provide an accurate estimate of hilltop curvature and thus, the sediment transport coefficient.

These data suggest that estimating $D$ from low resolution topographic data is possible in many landscapes, particularly those which have average ridgelines broader than the grid resolution of the topographic data. In the case of landscapes with sharper ridgelines such as Santa Cruz Island and the 750 Oregon Coast Range, it is more challenging to constrain $D$ effectively as the grid resolution is increased. The magnitude of overestimation of $D$ between the highest and lowest resolution diffusivity estimates, $0.0023\ m^2 a^{-1}$ in the case of the Oregon Coast range, will be a product of the uncertainty within the calculation of the erosion rate and material densities in addition to the local variations of $D$ within each landscape.

### 5.4 Hillslope length and relief

Measurements of hillslope length and relief have been used to test sediment flux laws (Roering et al., 2007; Grieve et al., 2016a) and to identify landscape transience (Hurst et al., 2013b; Mudd, 2016). Such analyses have previously been restricted to high resolution topographic data. When considering hillslope length, we must select a grid resolution that is at least half the median hillslope 760 length in order to resolve any useful information. However, in reality more than 2 pixels are required if any meaningful information is to be extracted from topographic data. As the median hillslope length for many landscapes has been shown to be in excess of 100 meters (Grieve et al., 2016a), this requirement for several pixels per hillslope falls well within the range of many lower resolution data products. Therefore, our results show that meaningful hillslope length measurements can be made 765 from lower resolution topographic data, with data products approaching 30 meter resolution proving suitable in some cases.

The relief measurements for each landscape, however, show more sensitivity to grid resolution, with a systematic increase in the median values in each location beyond 10 meters grid resolution. As



increasing grid size acts as a low-pass filter on the landscape, the elevation of ridges are expected to
be reduced, whilst the elevation of channel beds are raised, producing a net reduction in topographic
relief. However, the increased relief observed with increasing grid size is produced by the decrease
in drainage density with increasing grid size observed in Figure 6 producing fewer channels reaching
up towards ridgelines leading to hillslope flow paths traveling further downslope before reaching a
channel.

By contrasting the $L_H$ and $R$ results computed using fixed and variable channel heads, it is clear
that the optimal method for measuring hillslope length and relief is to employ as accurate a chan-
nel network as possible. However, the variable channel head data shows that the signal of average
hillslope length and relief is broadly insensitive to data resolution up to grid sizes of at least 10 me-
ters. This would facilitate the analysis of landscape transience using these measurements at a global
scale, using high resolution satellite derived DEMs, such as TanDEM-X (Krieger et al., 2007). This
relationship is again strongest in Gabilan Mesa, the landscape with the least topographic complexity
which demonstrates the least sensitivity to curvature measurements and the estimation of diffusiv-
ity. However, even in the more noisy landscape of the Oregon Coast Range, meaningful hillslope
length and relief measurements can still be made through the use of a geometric channel extraction
algorithm and lower resolution topographic data.

## 6 Conclusions

Through generation of topographic data spanning the range of grid resolutions currently used in
much of geomorphic research, a number of key metrics have been evaluated for their sensitivity to
grid resolution. We have demonstrated the reduction in the range of total and tangential curvature val-
ues as grid resolution is decreased, across three test landscapes. These curvature measurements are
important in the estimation of the hillslope sediment transport coefficient ($D$), in their use as a proxy
for erosion rate, and in the extraction of channel networks from topographic data. We demonstrate
that the estimation of $D$ from low resolution topographic data is possible, particularly in landscapes
such as Gabilan Mesa where hilltops are broad. Higher resolutions are required to extract meaningful
curvature information in steep landscapes with sharp ridges and narrow gullies.

The extraction of channel networks from digital topographic data is a significant challenge at all
spatial scales, as the definition of a channel network is integral in the execution of many analyses
(e.g., DiBiase et al., 2012; Hurst et al., 2012; Grieve et al., 2016a). We demonstrate that the use of
a geometric channel extraction algorithm produces channel networks for all three of our landscapes
which correspond well to networks extracted from high resolution topography. This correspondence
is tested through the computation of quality indexes for each predicted network, which outline the
suitability of this algorithm over a process based method at coarse DEM resolutions.



Average values of hillslope length and relief for each landscape are shown to be broadly insensitive to grid resolution up to grid sizes which correspond to the highest resolution topographic data globally available. This indicates that these measurements can be used to identify landscape transience in locations where LiDAR data are unavailable. The accuracy of these measurements is dependent on the accuracy of the channel network used, however, as using a geometric method of channel extraction from the 1 meter DEM still provides robust measurements of hillslope length and relief.

The relationships between decreasing grid resolution and the geomorphic parameters explored here demonstrate the influence of the spatial scale of the topographic expression of process on the quality of results which can be extracted from lower resolution topography. From these analyses it is challenging to identify a clear threshold below which data becomes unsuitable for use in geomorphic analysis. Rather, it is important to highlight the influence of landscape morphology and the dominant processes acting upon it in the selection of an appropriate data resolution for a study. Using this work as a framework, it is now possible to place constraints on the accuracy of results derived from coarse resolution topographic data, particularly in locations where no high resolution data are available.

*Acknowledgements.* The topographic data used in this study are freely available from http://www.OpenTopography. org, and the specific point clouds used can be downloaded from http://www.geos.ed.ac.uk/~s0675405/Res_ Data/Res_Data_Pack.zip. All of the code used in this analysis is open source and the topographic analysis routines are available at http://github.com/LSDtopotools/LSD_Resolution and the code to generate the figures in this paper, alongside the raw plot data can be downloaded from http://github.com/sgrieve/Resolution_Paper_ Figs. SMM and SWDG are funded by NERC grant NE/J009970/1 and SMM is funded by U.S. Army Research Office contract number W911NF-13-1-0478. FJC is funded by the Carnegie Foundation for the Universities of Scotland. DTM was funded by a NERC Doctoral Training Grant NE/152830X/1 and NE/J500021/1. DJF was funded by US National Science Foundation grant EAR-1420831. We thank Marie-Alice Harel for comments on an earlier version of the manuscript.

## Author contributions

SWDG, SMM, DTM and FJC wrote the software. SWDG performed the analysis. DJF and SMM resurrected the spectral filtering analysis from an unpublished 2002 manuscript, because they are lovers of the long game. SWDG wrote the paper with contributions from other authors.

## Appendix A: Channel extraction parameters

This table provides the parameters used to generate channel networks both using the geometric method and the DrEICH method. The drainage area value is used to thin the initial extracted network by removing channels which have a drainage area below the threshold value. The connected components value defines the point at which a group of contiguous channel pixels are considered to





be connected. The $\frac{m}{n}$ ratio is determined using software provided by Mudd et al. (2014) and its use within this context is discussed in detail in Clubb et al. (2014).

**Table A1.** Parameters used by the geometric and process based techniques in the extraction of channel networks.

| Location | Window radius $(m)$ | Drainage area $(m^2)$ | Connected components (Pixels) | $\frac{m}{n}$ ratio | Reference |
|---|---|---|---|---|---|
| Santa Cruz Island | 4 | 4 | 5 | 0.50 | This study |
| Gabilan Mesa | 5 | 4 | 5 | 0.45 | Grieve et al. (2016a, b) |
| Oregon Coast Range | 4 | 4 | 5 | 0.45 | Grieve et al. (2016a, b) |





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





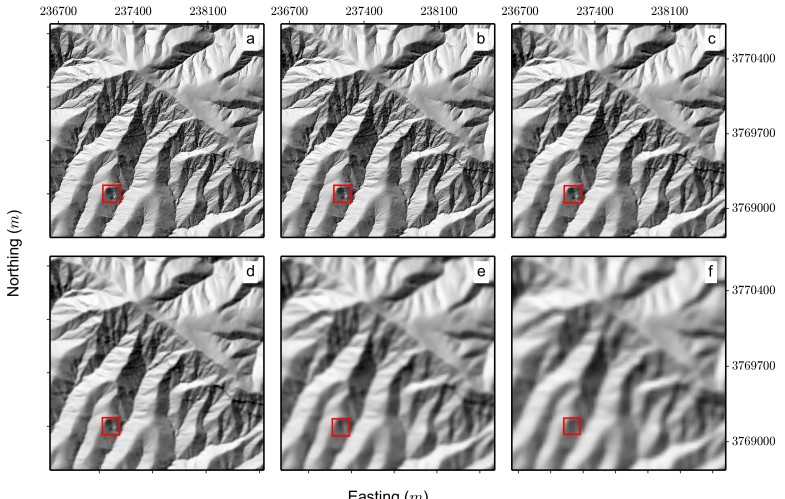

**Figure 1.** Example shaded reliefs of the same section of Santa Cruz Island at increasing grid resolutions. All coordinates are in UTM Zone 11N. Panels a to f represent resolutions of 1, 2, 5, 10, 20 and 30 meters. The red box outlines an extensively gullied first order drainage, clearly visible in the highest resolution data, but as the grid size is increased, this feature, and its internal structure becomes indistinguishable from the surrounding hillslopes.





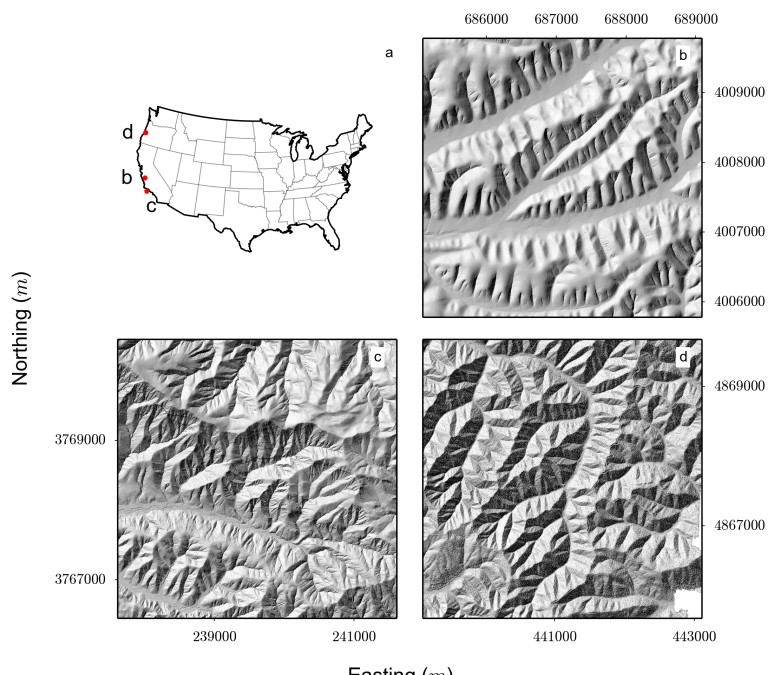

**Figure 2.** (a) Map showing the location of each of the study sites within the USA. (b-d) Shaded reliefs of representative sections of each study site, generated from 1 meter resolution data. All coordinates are in UTM. (b) Gabilan Mesa, California, UTM Zone 10N. (c) Santa Cruz Island, California, UTM Zone 11N. (d) Oregon Coast Range, Oregon, UTM Zone 10N.




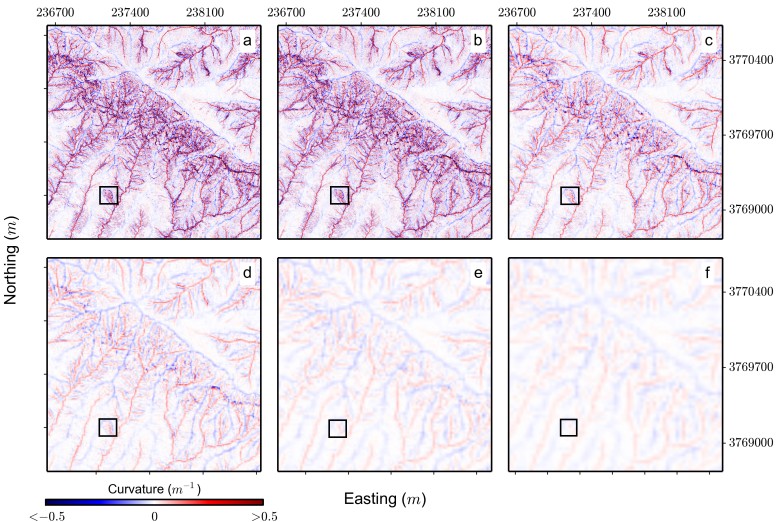

**Figure 3.** Maps showing the spatial variation in total curvature measurements as grid resolution is decreased for the same section of Santa Cruz Island as displayed in Figure 1. All coordinates are in UTM Zone 11N. Panels a to f represent resolutions of 1, 2, 5, 10, 20 and 30 meters. The black boxes outline the same features as highlighted in Figure 1, showing the reduction in the curvature signal with grid resolution for such a feature.





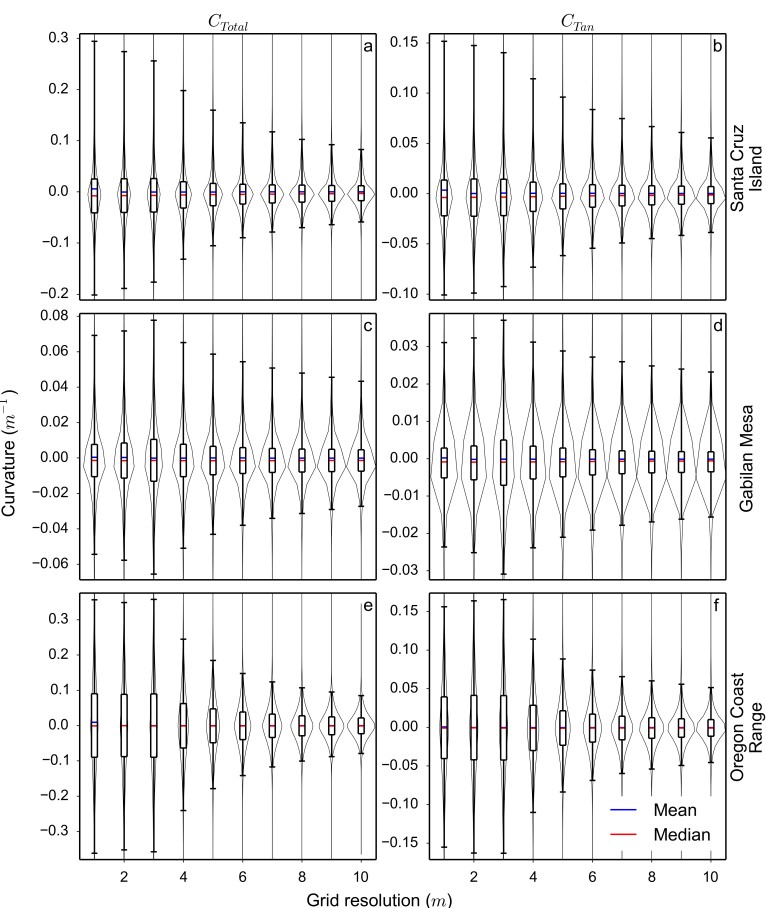

**Figure 4.** Plots of the distribution of $C_{Total}$ (a, c and e) and $C_{Tan}$ (b, d and f) measurements as resolution is decreased for each of the study landscapes. Whiskers are the 2$^{nd}$ and 98$^{th}$ percentiles, the box covers the 25$^{th}$ and 75$^{th}$ percentiles, the blue bar is the mean and the red bar is the median. The gray outline is the probability density function of each dataset.





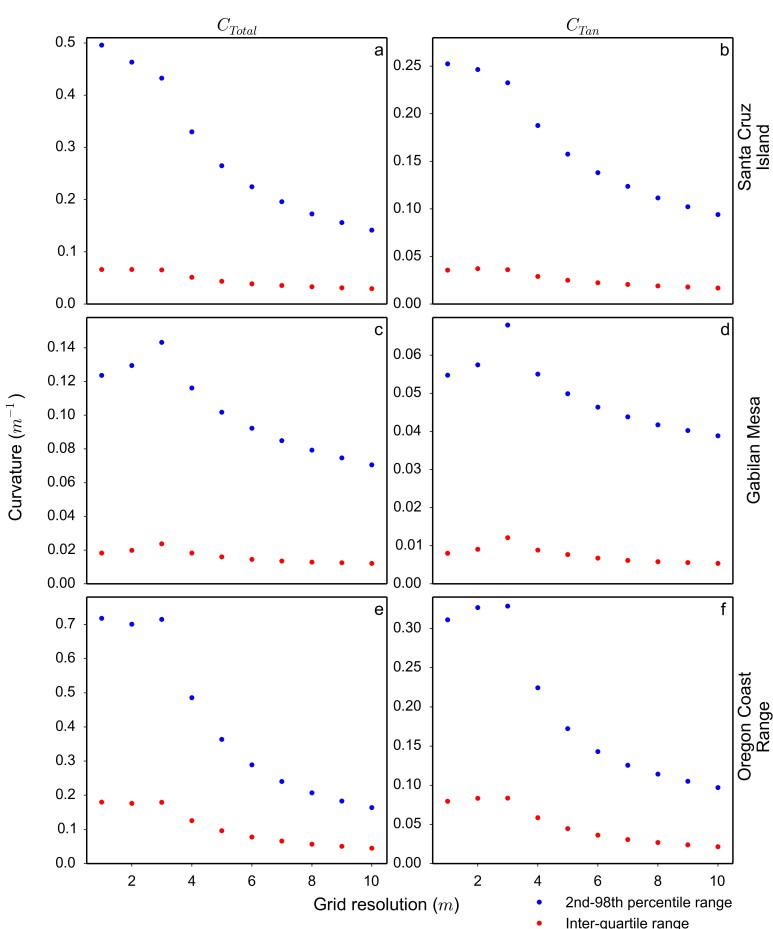

**Figure 5.** Plots of the reduction in range between the $2^{nd}$ and $98^{th}$ percentiles (blue) and the inter-quartile range (red) of $C_{Total}$ (a, c and e) and $C_{Tan}$ (b, d and f) measurements as resolution is decreased for each of the study landscapes.





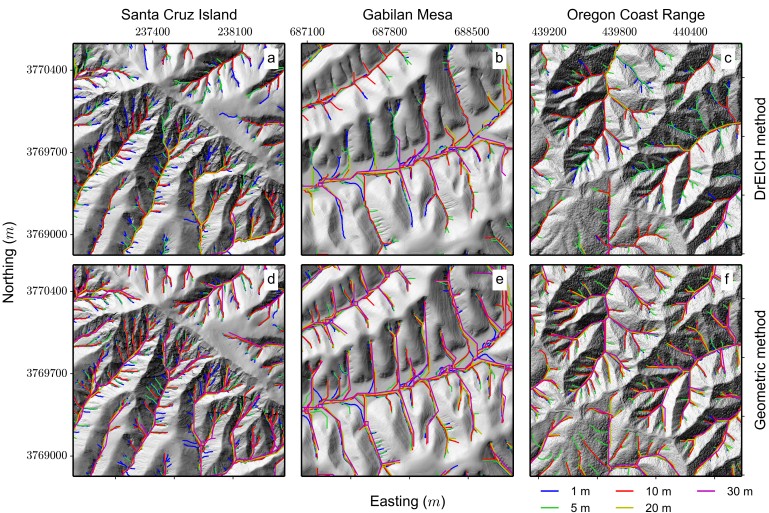

**Figure 6.** Representative sections of each landscape's channel network displaying the extent of each network as grid resolution is increased. Plots a, b and c are generated using the DrEICH method of channel extraction. Plots d, e and f are generated using the geometric method. All coordinates are in UTM. The left column is from Santa Cruz Island, UTM Zone 11N, the central column is from Gabilan Mesa, UTM Zone 10N and the right column is from the Oregon Coast Range, UTM Zone 10N.





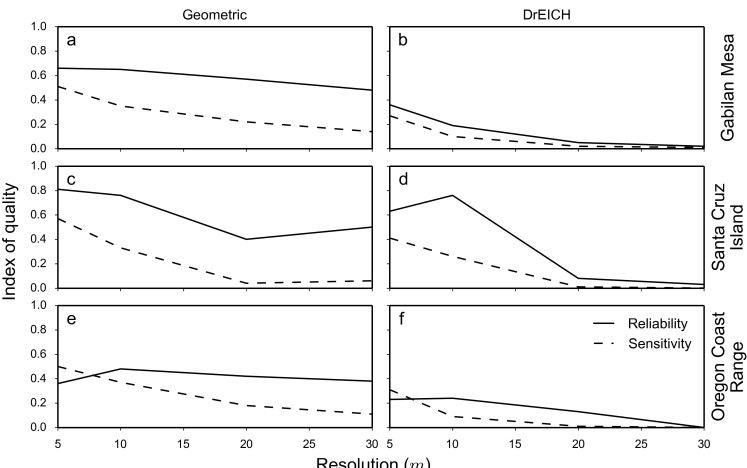

**Figure 7.** The variations in reliability (Equation 5) and sensitivity (Equation 6) of each channel network with decreasing grid resolution. Plots a, c and e are generated using the geometric method of channel extraction. Plots b, d and f are generated using the DrEICH method. The top row is from Gabilan Mesa, the middle row is from Santa Cruz Island and the bottom row is from the Oregon Coast Range. The full results from this analysis can be found in Tables 3 and 4.

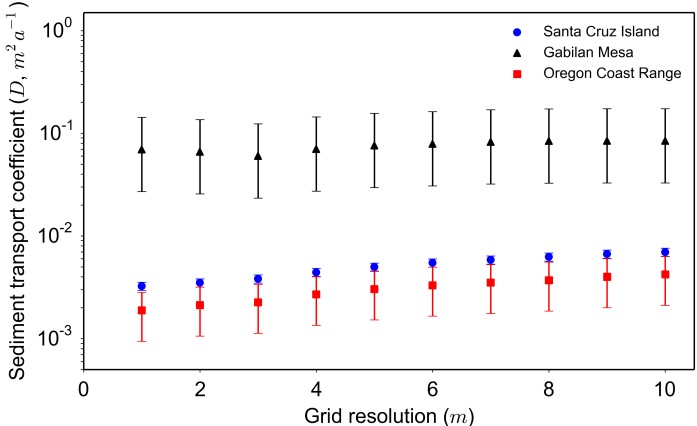

**Figure 8.** Changes in the estimated sediment transport coefficient, $D$, calculated using Equation 10 and parameters in Table 2 for each of the three study landscapes. The errorbars on each datapoint represent the uncertainties reported for each landscape's erosion rate data.





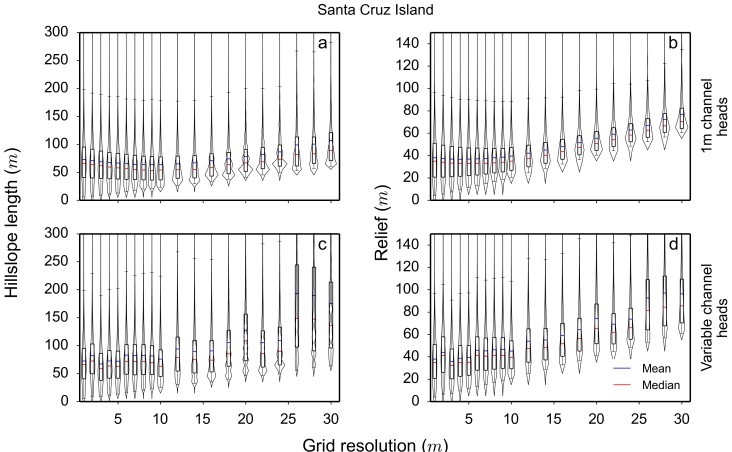

**Figure 9.** Plots of the distribution of hillslope length (a and c) and relief (b and d) measurements as resolution is decreased for Santa Cruz Island. Whiskers are the 2nd and 98th percentiles, the box covers the 25th and 75th percentiles, the blue bar is the mean and the red bar is the median. The gray outline is the probability density function of each dataset. The top row presents the best case scenario, where an independent constraint on the channel network is available for the lower resolution data and the bottom row uses the channel networks extracted using the geometric method outlined in Section 2.3 for each resolution step.



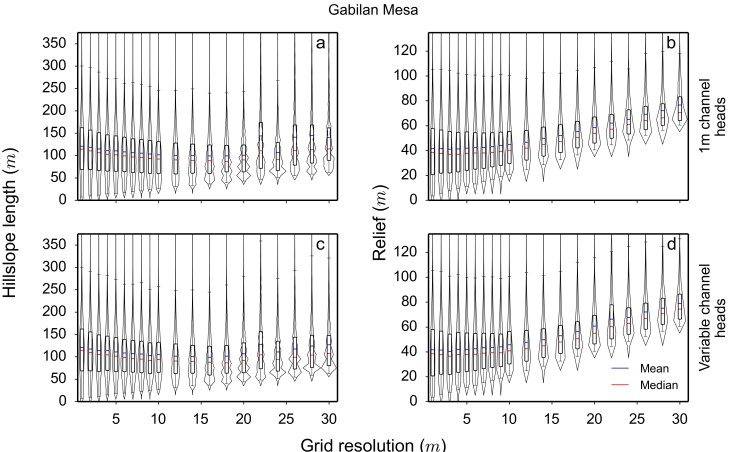

**Figure 10.** Plots of the distribution of hillslope length (a and c) and relief (b and d) measurements as resolution is decreased for Gabilan Mesa. Whiskers are the $2^{nd}$ and $98^{th}$ percentiles, the box covers the $25^{th}$ and $75^{th}$ percentiles, the blue bar is the mean and the red bar is the median. The gray outline is the probability density function of each dataset. The top row presents the best case scenario, where an independent constraint on the channel network is available for the lower resolution data and the bottom row uses the channel networks extracted using the geometric method outlined in Section 2.3 for each resolution step.





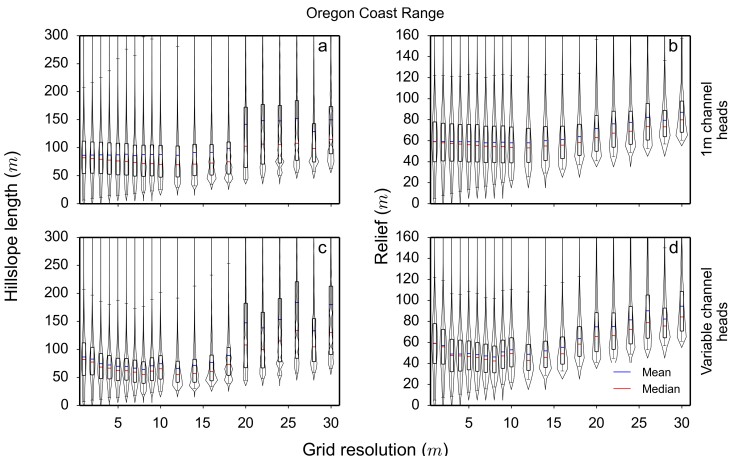

**Figure 11.** Plots of the distribution of hillslope length (a and c) and relief (b and d) measurements as resolution is decreased for the Oregon Coast Range. Whiskers are the 2$^{nd}$ and 98$^{th}$ percentiles, the box covers the 25$^{th}$ and 75$^{th}$ percentiles, the blue bar is the mean and the red bar is the median. The gray outline is the probability density function of each dataset. The top row presents the best case scenario, where an independent constraint on the channel network is available for the lower resolution data and the bottom row uses the channel networks extracted using the geometric method outlined in Section 2.3 for each resolution step. At higher resolution steps the 98$^{th}$ percentile data is not shown in the plot, to better highlight the distribution of measurements between the 25$^{th}$ and 75$^{th}$ percentiles, which make up the majority of the data points.





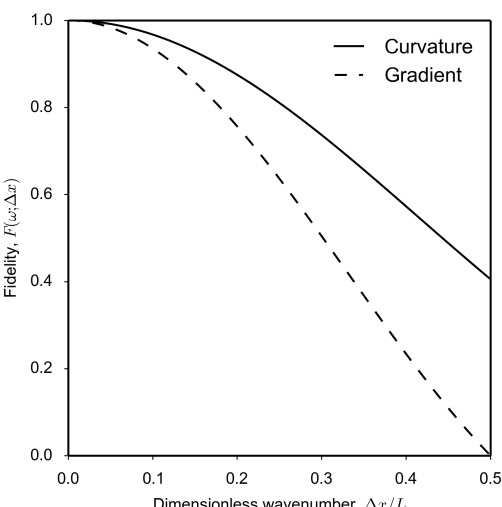

**Figure 12.** Plot of fidelity ($F$) of two one dimensional differencing operations: curvature (Equation 11) and to-
pographic gradient (Equation 15) as a function dimensionless wavenumber $\Delta x/L$ to the Nyquist wavenumber,
$\Delta x/L = 0.5$.



**Table 1.** LiDAR point cloud metadata.

| Location | Point density (points per $m^2$) | Vertical accuracy (m) | Horizontal accuracy (m) |
|---|---|---|---|
| Santa Cruz Island | 8.27 | $0.067^a$ | $1.07^a$ |
| Gabilan Mesa | 5.56 | $0.20 \pm 0.15$ | 0.11 |
| Oregon Coast Range | 6.55 | $0.07 \pm 0.03$ | 0.06 |

[a] Accuracy is the 95% confidence level of the root mean squared error of measurements compared to static GPS control points.

**Table 2.** Published parameters used to calculate diffusivity.

| Location | Soil density $(kgm^{-3})^a$ | Rock density $(kgm^{-3})^a$ | Erosion rate $(mmyr^{-1})$ | Reference |
|---|---|---|---|---|
| Santa Cruz Island | 1.4 | 2.4 | $0.069 \pm 0.007$ | Perroy et al. (2012) |
| Gabilan Mesa | 1.4 | 2.4 | $0.36^{+0.38}_{-0.22}$ | Roering et al. (2007) |
| Oregon Coast Range | 1.4 | 2.4 | $0.1 \pm 0.05$ | Roering et al. (1999) |

[a] Soil and rock densities representative of typical measurements of the fieldsites and are taken from Hillel (1980)

**Table 3.** Reliability and sensitivity metrics for the DrEICH method of channel extraction.

| Location | Resolution (m) | $\sum TP$ | $\sum FP$ | $\sum FN$ | $r$ | $s$ |
|---|---|---|---|---|---|---|
| Gabilan Mesa | 5 | 555 | 982 | 1489 | 0.36 | 0.27 |
| | 10 | 210 | 879 | 1875 | 0.19 | 0.1 |
| | 20 | 42 | 734 | 2088 | 0.05 | 0.02 |
| | 30 | 13 | 609 | 2122 | 0.02 | 0.01 |
| Santa Cruz Island | 5 | 3295 | 1971 | 4799 | 0.63 | 0.41 |
| | 10 | 2454 | 793 | 6865 | 0.76 | 0.26 |
| | 20 | 69 | 838 | 8235 | 0.08 | 0.01 |
| | 30 | 27 | 915 | 8284 | 0.03 | 0.0 |
| Oregon Coast Range | 5 | 507 | 1718 | 1131 | 0.23 | 0.31 |
| | 10 | 144 | 445 | 1462 | 0.24 | 0.09 |
| | 20 | 16 | 105 | 1623 | 0.13 | 0.01 |
| | 30 | 2 | 442 | 1639 | 0.0 | 0.0 |



**Table 4.** Reliability and sensitivity metrics for the geometric method of channel extraction.

| Location | Resolution (m) | $\sum TP$ | $\sum FP$ | $\sum FN$ | $r$ | $s$ |
|---|---|---|---|---|---|---|
| Gabilan Mesa | 5 | 1019 | 519 | 987 | 0.66 | 0.51 |
| | 10 | 712 | 380 | 1301 | 0.65 | 0.35 |
| | 20 | 448 | 332 | 1592 | 0.57 | 0.22 |
| | 30 | 292 | 333 | 1775 | 0.48 | 0.14 |
| Santa Cruz Island | 5 | 4280 | 991 | 3109 | 0.81 | 0.57 |
| | 10 | 2473 | 777 | 4998 | 0.76 | 0.33 |
| | 20 | 334 | 505 | 7861 | 0.4 | 0.04 |
| | 30 | 475 | 470 | 7659 | 0.5 | 0.06 |
| Oregon Coast Range | 5 | 792 | 1438 | 788 | 0.36 | 0.5 |
| | 10 | 562 | 602 | 938 | 0.48 | 0.37 |
| | 20 | 276 | 374 | 1275 | 0.42 | 0.18 |
| | 30 | 475 | 277 | 1418 | 0.38 | 0.11 |