# Peer review of "How does grid-resolution modulate the topographic expression of geomorphic processes?"

_Earth Surface Dynamics, 2016_

## Referee Comment (RC1) · Stuart W. D. Grieve et al. · 30 May 2016

This manuscript seeks to quantify how well topographic information of varying resolution can be used to extract quantitative information for landscape evolution interpretation and simulation. The authors have crafted a clear goal in this respect and chosen three well-trod soil-mantled landscapes to explore how successive degradation of DEM resolution affects estimates of curvature, slope length, relief, and channel network extent. While the geomorphic community trumpets lidar's superior capabilities for these endeavors, the authors are correct in noting that the extent of lidar coverage is limiting (although we should still advocate for global coverage!). This manuscript succeeds because it nicely presents empirical findings and because it incorporates a nifty theoretical explanation for why we lose the ability to resolve features of a given wavelength

with changing grid spacing. Below, I describe a small handful of suggestions that should be easily incorporated if so chosen.

Major comments: 1) Are DEMs of the same grid spacing necessarily equal? The issue of varying DEM grid spacing is the fundamental message of this manuscript and the authors argue for the utility of the results obtained here for interpreting TanDEM-X and SRTM data, but these datasets are derived differently from lidar data and it's not clear to me that the exercise laid out here will always be relevant in this respect. In other words, can the authors show that degrading lidar data to 12meters is essentially the same as obtaining a TanDEM-X dataset? Or similarly for a 30-m SRTM dataset? Because we hold up lidar as the gold standard, it's appropriate as the reference but it seems that additional factors will likely affect how well radar-based techniques can resolve surface features. It would be highly convincing, for example, if the authors could show that a degraded lidar dataset is consistent with TanDEM-X or SRTM data in a simple way that closes the loop and truly opens the door for widespread use of the principles established in this very fine paper. Perhaps the answer is already known and a simple "yes, this works" can be garnered from the literature.

2) Language: the manuscript cycles through the terms "resolution" and "grid spacing" in various forms and there are a few places where the meaning can become confusing (e.g., line 497). The word "resolution" is sometimes traded for "spacing", inviting ambiguity and my suggestion is to simplify wherever possible.

3) Broadly convex hilltops can be resolved with coarse data. The authors nicely show that curvature at the Gabilan Mesa site is retained as grid spacing increases, which is a very nice result. After seeing more and more of the GM lidar data, it does appear that many sections of that study area exhibit remnant surfaces that are not yet adjusted to regional baselevel lowering, which is a potential bias that could merit mention. That said, I think the result here is nonetheless robust. perhaps more importantly, though, I wonder how this result would be useful for a researcher working in a far-flung field area without a priori knowledge of whether their study site exhibits broadly convex hilltops.

[Figure]

How would they be able to determine whether they're working with an Oregon Coast Range or Gabilan Mesa like study area? How will they know to trust their curvature values or not?

4) Theoretical explanation for grid spacing dependency on feature resolution. Section 5.1.1. does a very nice job of laying out a spectral analysis-based explanation for this problem. It nicely complements the empirical analysis here but it is terse and challenging to follow. My concern is that readers without a background in signal processing will not be able to follow it. The wonderful outcome in lines 640s to 660s merits much fanfare and is well-stated, but getting there could be much more straightforward. On line 629-634, exactly how does the gain equation show the associations stated, for example? There is no curvature value in this equation (strictly speaking), so the connection b/w gain and the traditional metrics could be more clear. In fact, my sense is that this section would be more effective as a substantial part of the manuscript methodology rather than a theoretical afterthought stuck in the discussion. This analysis is central to the message here and the authors might consider putting it upfront because it bears on all of the empirical results. That said, it would require some hand holding to be effective in that role. In particular, the goal needs to be clearly stated as well as simply stated explanations about wavenumber response function, continuous vs. discontinuous waveforms, and gain. The explanation for fidelity is nicely stated but I fear most readers will have to do substantial work to get there...which would be rewarded if they do so, however! A major rehaul is not the suggestion, but rather a more integrated role for this section. In particular, the very different curvature and slope fidelity curves are highly compelling (fig 12). Because the loss of slope fidelity explains previously published papers, this also strikes me as something that's more meaty than discussion subsection material.

Specifics: The text is very well written with excellent figures, too.

---

## Referee Comment (RC2) · Anonymous Referee #2 · 7 Jun 2016

The purpose of this work was to analyze the effectiveness of lower resolution topographic data to understand Earth surface processes. In detail, the relationship between curvature and grid resolution is considered, alongside the estimation of the hillslope sediment transport coefficient for each study area. The results suggested that although high resolution (e.g., 1 m) topographic data does yield exciting possibilities for geomorphic research, many key parameters can be understood in lower resolution data, given careful consideration of how analyses are performed.

The paper is interesting. Even if we are living in the "high-resolution topography age", still we can obtain benefits (in term of understanding Earth surface processes) from low-resolution topographic information.

[Figure]

**ESurfD**
However, the reason to work with low-resolution data is not only because, as author stated, global lidar coverage cannot be achieved in the near future (I'm quite optimistic for the future, technology is evolving very fast and big data is one of the major challenges for this century....). I believe that one reason is also because, with larger grid cell size, we can better represent the scale at which few processes occur. I suggest to highlight this in the text; the paper will be benefited from such discussion. I suggest also to read the work of Tarolli and Tarboton (2006), where it was found that, the slope calculated with 10 m DTM (from lidar) allowed a better performance of the shallow landslide model they used. The slope calculated with 2 m DTM was not representative of the scale at which the analyzed shallow landslides occurred. Digital terrain model scales larger than 10 m result in loss of resolution that degrades the results, while for digital terrain model scales smaller than 10 m the physical processes responsible for triggering landslides are obscured by smaller scale terrain variability.

The results of this work are in line with such findings: it is possible to estimate suitable sediment transport coefficients also from low-resolution topographic data. The paper is clear and it merits to be published.

I just suggest just few minor changes: - Improve a little the discussion on the grid cell size and the scale at which a physical process occur. - Fig.1,2,3,6: add the scale bar.

Reference Tarolli, P., Tarboton, D.G., (2006). A New Method for Determination of Most Likely Landslide Initiation Points and the Evaluation of Digital Terrain Model Scale in Terrain Stability Mapping, Hydrol. Earth Syst. Sci., 10, 663-677, doi:10.5194/hess-10-663-2006.

---

## Author Comment (AC1) · 8 Jun 2016

Throughout this document the reviewer's comments are in **bold type** and our responses are in standard type.

**This manuscript seeks to quantify how well topographic information of varying resolution can be used to extract quantitative information for landscape evolution interpretation and simulation. The authors have crafted a clear goal in this respect and chosen three well-trod soil-mantled landscapes to explore how successive degradation of DEM resolution affects estimates of curvature, slope length, relief, and channel network extent. While the geomorphic community**

**trumpets lidar's superior capabilities for these endeavors, the authors are correct in noting that the extent of lidar coverage is limiting (although we should still advocate for global coverage!). This manuscript succeeds because it nicely presents empirical findings and because it incorporates a nifty theoretical explanation for why we lose the ability to resolve features of a given wavelength with changing grid spacing. Below, I describe a small handful of suggestions that should be easily incorporated if so chosen.**

We thank the reviewer for their detailed and positive consideration of our work. We are delighted that the reviewer considers the manuscript to be successful in exploring the influence of grid resolution on geomorphic analysis. We believe that through the discussion points highlighted below, the manuscript has been significantly enhanced, and will consequently be of greater value to a wider audience. At the recommendation of the reviewer we have expanded the introduction to highlight the challenges inherent in comparing different data products, with the aim of giving readers more confidence in the use of lower resolution data products, where appropriate. We have clarified the terminology used throughout the manuscript and highlighted minor points which enhance the interpretation of our curvature results. The spectral analysis section has been expanded to provide a clearer explanation of the equations used, in order to better highlight the relevance of this section to the remainder of the manuscript.

**Major comments: 1) Are DEMs of the same grid spacing necessarily equal? The issue of varying DEM grid spacing is the fundamental message of this manuscript and the authors argue for the utility of the results obtained here for interpreting TanDEM-X and SRTM data, but these datasets are derived differently from lidar data and it's not clear to me that the exercise laid out here will always be relevant in this respect. In other words, can the authors show that degrading lidar data to 12meters is essentially the same as obtaining a**

**TanDEM-X dataset? Or similarly for a 30-m SRTM dataset? Because we hold up lidar as the gold standard, it's appropriate as the reference but it seems that additional factors will likely affect how well radar-based techniques can resolve surface features. It would be highly convincing, for example, if the authors could show that a degraded lidar dataset is consistent with TanDEM-X or SRTM data in a simple way that closes the loop and truly opens the door for widespread use of the principles established in this very fine paper. Perhaps the answer is already known and a simple 'yes, this works' can be garnered from the literature.**

This direct comparison between data products was something we considered during the planning of this research. We decided to avoid directly testing particular data products as we wanted to show the effect of changes in data resolution, rather than provide a quality estimate for a range of data sources. We agree that such a comparison would be a compelling result to add to this work, however, a full evaluation of the similarity between data products would be beyond the scope of this contribution. The errors between LiDAR data and other data products are region specific, with errors appearing to be connected to landscape morphology and it would be challenging to make a broad conclusion that any SRTM or TanDEM-X data product is globally equivalent to down-sampled LiDAR without local scale testing. We have added a brief discussion of these challenges, highlighting the need for further evaluation of these datasets to Section 2.1.

**2) Language: the manuscript cycles through the terms 'resolution' and 'grid spacing' in various forms and there are a few places where the meaning can become confusing (e.g., line 497). The word 'resolution' is sometimes traded for 'spacing', inviting ambiguity and my suggestion is to simplify wherever possible.**

We have gone through the manuscript and clarified our terminology, to only refer to

increasing or decreasing (grid) resolution, rather than grid sizes or spacing.

**3) Broadly convex hilltops can be resolved with coarse data. The authors
nicely show that curvature at the Gabilan Mesa site is retained as grid spacing
increases, which is a very nice result. After seeing more and more of the GM
lidar data, it does appear that many sections of that study area exhibit remnant
surfaces that are not yet adjusted to regional baselevel lowering, which is a
potential bias that could merit mention. That said, I think the result here is
nonetheless robust.**

We have added a statement into Section 3.1 to highlight this complexity within Gabilan
Mesa.

**perhaps more importantly, though, I wonder how this result would be useful
for a researcher working in a far-flung field area without a priori knowledge
of whether their study site exhibits broadly convex hilltops. How would they
be able to determine whether they're working with an Oregon Coast Range or
Gabilan Mesa like study area? How will they know to trust their curvature values
or not?**

This will always remain a challenge inherent in desk based research, and if the
only data available to a researcher is low resolution topographic data, it would be
challenging to be entirely confident in measurements of hilltop curvature. We have
added a statement to the discussion (Section 5.1.2) to emphasize this point, although
in most cases it would be possible to gain a qualitative understanding of the landscape
morphology through other non-topographic datasets.

**4) Theoretical explanation for grid spacing dependency on feature resolution. Section 5.1.1. does a very nice job of laying out a spectral analysis-based explanation for this problem. It nicely complements the empirical analysis here but it is terse and challenging to follow. My concern is that readers without a background in signal processing will not be able to follow it. The wonderful outcome in lines 640s to 660s merits much fanfare and is well-stated, but getting there could be much more straightforward. On line 629-634, exactly how does the gain equation show the associations stated, for example? There is no curvature value in this equation (strictly speaking), so the connection b/w gain and the traditional metrics could be more clear. In fact, my sense is that this section would be more effective as a substantial part of the manuscript methodology rather than a theoretical afterthought stuck in the discussion. This analysis is central to the message here and the authors might consider putting it upfront because it bears on all of the empirical results. That said, it would require some hand holding to be effective in that role. In particular, the goal needs to be clearly stated as well as simply stated explanations about wavenumber response function, continuous vs. discontinuous waveforms, and gain. The explanation for fidelity is nicely stated but I fear most readers will have to do substantial work to get there...which would be rewarded if they do so, however! A major rehaul is not the suggestion, but rather a more integrated role for this section. In particular, the very different curvature and slope fidelity curves are highly compelling (fig 12). Because the loss of slope fidelity explains previously published papers, this also strikes me as something that's more meaty than discussion subsection material.**

We thank the reviewer for their positive comments about this section. Prior to submission, we debated placing this section in the methods section rather than within the results section. We felt that this led to a more disjointed narrative than is present in the current structure of the manuscript. In other words, we tried what the

reviewer suggests and didn't really like the result: the manuscript flows better if the numerical problem is highlighted and then we use the analytical solutions to show why this problem exists. This ensures that the paper's focus is on the practical problem of determining what one can actually say about a low resolution DEM rather than throwing the reader into a cold bath of spectral analysis before any results are reported.

We will follow the reviewer's advice on expanding this section so it becomes (we hope) clearer to readers where the equations have come from and what they mean.

---

## Referee Comment (RC3) · W. Schwanghart (Referee) · 9 Jun 2016

Grieve et al.'s study investigates how different spatial resolutions of digital elevation models (DEMs) affect topographic derivatives that are particularly relevant for characterizing geomorphological processes. They place emphasis on the second derivative curvature that is the basis for channel network identification, estimating hillslope diffusivity, and measuring hillslope length and relief. I have rarely received a request for reviewing a paper that is in such good shape. The manuscript is very well written, concise, and the methodology sound. Overall, the conclusions drawn by the authors are well supported by their analysis. Notwithstanding, I have two comments that should be addressed before the manuscript is ready to be published.

1. The problem of coarsening resolutions is addressed by downsampling high-resolution LiDAR data. However, this approach neglects that DEMs are acquired by different sensors that may generate artefacts due to vegetation, shadowing, foreshort-ening, etc. These systematic data errors are likely not captured by the local binning algorithm that the authors used to downsample the dense point clouds. My concern is that, now that a globally available DEM with 12 m resolution is available (WorldDEM), researchers may place a possibly to high confidence into the fidelity of that product. The incorporation of this data (or other data sources) into the analysis would provide guidance here.

2. The mathematical treatment of the observed loss of fidelity with increasing spatial resolution appears somewhat misplaced in the discussion. Instead, this part could well serve as a motivation of the study that should be placed at the beginning of section 2. Moreover, I think that this part is not intelligible for many who are infamiliar with wavenumber response functions. Adding more detail here will certainly be thanked by the general readership of ESURF.

---

## Short Comment (SC1) · 14 Jun 2016

General comments This paper seeks to determine whether low-resolution (i.e., > 10 m grid cells) can be used to quantify topography relevant to geomorphic processes (channelization, hillslope diffusion, etc.). The authors document the grid-resolution dependence on the median values of curvature, slope, and relief, and on the fidelity of channel head identification algorithms. Their data demonstrates how decreasing grid resolution cuts off extreme values of topographic metrics, a finding well-represented in the literature but never so comprehensively. To explain this effect, they use spectral analysis to show why this effect occurs, and on the basis of this finding, argue that the utility of low-resolution data is highly dependent on the morphology of the study

landscape. This argument provides a promising way forward and gives hope for studies based on low-resolution data in landscapes with relatively long hillslopes (landscapes that support much of the human population). The paper is exceptionally well-written and organized; I have a few ideas I would like the authors to consider and a smattering of technical notes that will hopefully improve the clarity of the paper even further.

Specific comments 1.    In section 2.1, I would like to see more discussion/acknowledgment of or grappling with the issue of gridding point cloud data and potential over-interpolation of Lidar. For example, in the Oregon Coast Rang, forests are generally logging company plantings and have exceptionally high canopy density, occasionally limiting bare earth data to a point or two per hillslope, especially on steeper slopes. 2. The spectral analysis discussion (section 5.1) comes out of nowhere in the context of the paper's organization – it's not mentioned at all in the introduction, abstract or methods. Explaining the origin of the grid resolution effect is one of the great strengths of the paper; hence, I would advise more emphasis on these ideas – perhaps a section in the methods or theoretical underpinnings? 3. As noted, I like that the authors provide guidance for a way forward, but I take issue with their concluding assertion on lines 815 – 817. As presented in the paper, constraining the accuracy of coarse resolution results requires having high-resolution data to compare it to, or at the very least, the ability to measure hillslope length (which requires a lot of fieldwork, or high-resolution topography).

Technical notes - I'm interested to see what a log-scale on Figure 9 would look like. It seems like all the distributions are skewed and with a log scale we could maybe see more structure around the median value. - The points in Figure 5 are hard to see. - Section 1.1 labeling is superfluous as there is no section 1.2

---

## Author Comment (AC2) · 15 Jun 2016

Throughout this document the reviewer's comments are in **bold type** and our responses are in standard type.

**The purpose of this work was to analyze the effectiveness of lower resolution topographic data to understand Earth surface processes. In detail, the relationship between curvature and grid resolution is considered, alongside the estimation of the hillslope sediment transport coefficient for each study area. The results suggested that although high resolution (e.g., 1 m) topographic data does yield exciting possibilities for geomorphic research, many key parameters**

**can be understood in lower resolution data, given careful consideration of how analyses are performed. The paper is interesting. Even if we are living in the "high-resolution topography age", still we can obtain benefits (in term of understanding Earth surface processes) from low-resolution topographic information.**

We are pleased that the reviewer found the manuscript interesting and agree that even in this age of increased access to high resolution topography there is a lot of value in lower resolution data products. We hope that this contribution will assist in highlighting the utility of such datasets, as we continue to bridge the gap between locations with high resolution topography and those without. In responding to this review we have expanded the introduction to highlight that low resolution or downsampled high resolution data has utility as it can better capture processes which occur over larger length scales. We have also clarified the figure captions of a number of figures in order to ensure readers understand the spatial scales employed in visualizing topographic data. In making these changes we believe that the manuscript has the potential to reach a wider audience and has been enhanced by the comments contained in this review.

**However, the reason to work with low-resolution data is not only because, as author stated, global lidar coverage cannot be achieved in the near future (I'm quite optimistic for the future, technology is evolving very fast and big data is one of the major challenges for this century. . ..). I believe that one reason is also because, with larger grid cell size, we can better represent the scale at which few processes occur. I suggest to highlight this in the text; the paper will be benefited from such discussion. I suggest also to read the work of Tarolli and Tarboton (2006), where it was found that, the slope calculated with 10 m DTM (from lidar) allowed a better performance of the shallow landslide model**

**they used. The slope calculated with 2 m DTM was not representative of the scale at which the analyzed shallow landslides occurred. Digital terrain model scales larger than 10 m result in loss of resolution that degrades the results, while for digital terrain model scales smaller than 10 m the physical processes responsible for triggering landslides are obscured by smaller scale terrain variability**

We neglected to cover this concept fully within our original manuscript, several allusions were made to the scale dependence of processes and landforms, yet this idea was not clearly stated as a distinct point in the manuscript. We have added a section within the introduction explicitly highlighting this alternative use for low resolution data, using the suggested reference as a case study and tying it in to ideas of selecting window sizes for appropriate spatial averaging of topographic metrics.

**The results of this work are in line with such findings: it is possible to estimate suitable sediment transport coefficients also from low-resolution topographic data. The paper is clear and it merits to be published.**

**I just suggest just few minor changes: - Improve a little the discussion on the grid cell size and the scale at which a physical process occur.**

We have extended the introduction to clarify this point. See the response above for more detail.

**Fig.1,2,3,6: add the scale bar**

These figures are in UTM coordinates and so the axes act as the scale bar. We have not altered the figures, but have clarified this in each figure caption, to ensure that readers understand the scales at which we are representing spatial data.

---

## Author Comment (AC3) · 15 Jun 2016

Throughout this document the reviewer's comments are in **bold type** and our responses are in standard type.

**Grieve et al.'s study investigates how different spatial resolutions of digital elevation models (DEMs) affect topographic derivatives that are particularly relevant for characterizing geomorphological processes. They place emphasis on the second derivative curvature that is the basis for channel network identification, estimating hillslope diffusivity, and measuring hillslope length and relief. I have rarely received a request for reviewing a paper that is in such**

**good shape. The manuscript is very well written, concise, and the methodology sound. Overall, the conclusions drawn by the authors are well supported by their analysis. Notwithstanding, I have two comments that should be addressed before the manuscript is ready to be published.**

We thank the reviewer for such a positive appraisal of our work and we are delighted that the reviewer considers our work to be of a high standard. In addressing the two comments made below we believe that the manuscript has been enhanced, and hope that following the changes outlined below this research will be of greater value to a wider audience.

**1. The problem of coarsening resolutions is addressed by downsampling high resolution LiDAR data. However, this approach neglects that DEMs are acquired by different sensors that may generate artefacts due to vegetation, shadowing, foreshortening, etc. These systematic data errors are likely not captured by the local binning algorithm that the authors used to downsample the dense point clouds. My concern is that, now that a globally available DEM with 12 m resolution is available (WorldDEM), researchers may place a possibly to high confidence into the fidelity of that product. The incorporation of this data (or other data sources) into the analysis would provide guidance here.**

A similar observation was made by reviewer 1, regarding our ability to compare datasets derived using differing data collection and processing methods. We had initially considered performing such an analysis between differing data products, however, we elected to use downsampled LiDAR data as we wanted to the impact of changes solely in data resolution, isolating potential sources of error identified above, and allowing us to tie our results with the theoretical observations presented in Section 5.1.1. We have added further discussion of these issues and our motivation for the

use of LiDAR data to section 2.1, with the aim of ensuring that our work is considered within the proper context.

**2. The mathematical treatment of the observed loss of fidelity with increasing spatial resolution appears somewhat misplaced in the discussion. Instead, this part could well serve as a motivation of the study that should be placed at the beginning of section 2.**

A similar comment was made by reviewer 1 and our response is copied below:

Prior to submission, we debated placing this section in the methods section rather than within the results section. We felt that this led to a more disjointed narrative than is present in the current structure of the manuscript. In other words, we tried what the reviewer suggests and didn't really like the result: the manuscript flows better if the numerical problem is highlighted and then we use the analytical solutions to show why this problem exists. This ensures that the paper's focus is on the practical problem of determining what one can actually say about a low resolution DEM rather than throwing the reader into a cold bath of spectral analysis before any results are reported.

**Moreover, I think that this part is not intelligible for many who are infamiliar with wavenumber response functions. Adding more detail here will certainly be thanked by the general readership of ESURF.**

We will expand this section to better clarify our use of wavenumber response functions and tie the analysis more closely to the real-world results, with the aim of making this section more accessible to the general readership of ESURF.

**ESurfD**

Interactive
comment

---

## Author Comment (AC4) · 15 Jun 2016

Throughout this document the reviewer's comments are in **bold type** and our responses are in standard type.

**General comments This paper seeks to determine whether low-resolution (i.e., > 10 m grid cells) can be used to quantify topography relevant to geomorphic processes (channelization, hillslope diffusion, etc.). The authors document the grid-resolution dependence on the median values of curvature, slope, and relief, and on the fidelity of channel head identification algorithms. Their data demonstrates how decreasing grid resolution cuts off extreme values of**

[Figure]

**topographic metrics, a finding well-represented in the literature but never so comprehensively. To explain this effect, they use spectral analysis to show why this effect occurs, and on the basis of this finding, argue that the utility of low-resolution data is highly dependent on the morphology of the study landscape. This argument provides a promising way forward and gives hope for studies based on low-resolution data in landscapes with relatively long hillslopes (landscapes that support much of the human population). The paper is exceptionally well-written and organized; I have a few ideas I would like the authors to consider and a smattering of technical notes that will hopefully improve the clarity of the paper even further.**

We thank the reviewer for their thorough and positive appraisal of our work. We appreciate the constructive comments and have addressed them fully below. As a consequence of these comments we have expanded sections of the introduction, discussion and conclusion in addition to improving the clarity of one of the figures. We believe that as a result of these suggestions the manuscript is considerably stronger and will reach a wider audience.

**Specific comments 1. In section 2.1, I would like to see more discussion/acknowledgment of or grappling with the issue of gridding point cloud data and potential over-interpolation of Lidar. For example, in the Oregon Coast Range, forests are generally logging company plantings and have exceptionally high canopy density, occasionally limiting bare earth data to a point or two per hillslope, especially on steeper slopes.**

We have expanded the discussion in Section 2.1 around the gridding of LiDAR data to reflect this challenge, highlighting the difficulty of generating a high density of ground returns in heavily vegetated areas such as the Oregon Coast Range and justifying our

gridding of each dataset to 1 meter resolution with reference to previous studies.

**2. The spectral analysis discussion (section 5.1) comes out of nowhere in the context of the paper's organization - it's not mentioned at all in the introduction, abstract or methods. Explaining the origin of the grid resolution effect is one of the great strengths of the paper; hence, I would advise more emphasis on these ideas - perhaps a section in the methods or theoretical underpinnings?**

We have addressed this issue in response to reviewers 1 and 3 and will expand this section to provide more clarity to readers who are less familiar with these techniques, in addition to better relating the material in Section 5.1 to the rest of the manuscript's results. As recommended we will also add a more explicit statement of the spectral analysis within the abstract and introduction.

**3. As noted, I like that the authors provide guidance for a way forward, but I take issue with their concluding assertion on lines 815 - 817. As presented in the paper, constraining the accuracy of coarse resolution results requires having high-resolution data to compare it to, or at the very least, the ability to measure hillslope length (which requires a lot of fieldwork, or high-resolution topography).**

In responding to reviewer 1, we have provided more nuance to this idea within the discussion (Section 5.1.2), highlighting that some ancillary data or field exploration will be required in order to evaluate results derived from low resolution data. We have also revised lines 815-817 to reflect this more nuanced conclusion.

**Technical notes - I'm interested to see what a log-scale on Figure 9 would look**

**like. It seems like all the distributions are skewed and with a log scale we could maybe see more structure around the median value.**

Included in this response is the data in Figure 9 re-plotted using a log scale on the y-axis. It shows little change in the overall patterns with no structure around the median value becoming apparent in any of the 3 hillslope length and relief figures.

**The points in Figure 5 are hard to see.**

We have replotted this data using increased marker sizes, and have used differing marker shapes in addition to colors to better differentiate between the two datasets.

**Section 1.1 labeling is superfluous as there is no section 1.2**

This section labeling is a function of the Esurf latex template and should be resolved in the final form of the manuscript.
* * *
[Figure]

**Fig. 1.**